# Semantic Visual Anomaly Detection and Reasoning in AI-Generated Images

**Chuangchuang Tan**[1,2,3]*, **Xiang Ming**[3], **Jinglu Wang**[3], **Renshuai Tao**[1,2], **Bin Li**[4], **Yunchao Wei**[1,2], **Yao Zhao**[1,2]†, **Yan Lu**[3]†

[1]Institute of Information Science, Beijing Jiaotong University
[2]Visual Intellgence +X International Cooperation Joint Laboratory of MOE
[3]Microsoft Research Asia    [4]Shenzhen University
`tanchuangchuang@bjtu.edu.cn, yzhao@bjtu.edu.cn, yanlu@microsoft.com`

## Abstract

The rapid advancement of AI-generated content (AIGC) has enabled the synthesis of visually convincing images; however, many such outputs exhibit subtle **semantic anomalies**, including unrealistic object configurations, violations of physical laws, or commonsense inconsistencies, which compromise the overall plausibility of the generated scenes. Detecting these semantic-level anomalies is essential for assessing the trustworthiness of AIGC media, especially in AIGC image analysis, explainable deepfake detection and semantic authenticity assessment. In this paper, we formalize **semantic anomaly detection and reasoning** for AIGC images and introduce **AnomReason**, a large-scale benchmark with structured annotations as quadruples *(Name, Phenomenon, Reasoning, Severity)*. Annotations are produced by a modular multi-agent pipeline (**AnomAgent**) with lightweight human-in-the-loop verification, enabling scale while preserving quality. At construction time, AnomAgent processed approximately 4.17 B GPT-4o tokens, providing scale evidence for the resulting structured annotations. We further show that models fine-tuned on AnomReason achieve consistent gains over strong vision-language baselines under our proposed semantic matching metric (*SemAP* and *SemF1*). Applications to explainable deepfake detection and semantic reasonableness assessment of image generators demonstrate practical utility. In summary, AnomReason and AnomAgent serve as a foundation for measuring and improving the semantic plausibility of AI-generated images. We will release code, metrics, data, and task-aligned models to support reproducible research on semantic authenticity and interpretable AIGC forensics. The code is available at `https://github.com/chuangchuangtan/Semantic-Visual-Anomaly-Detection-and-Reasoning`.

## 1 Introduction

The rapid advancement of AI-generated content (AIGC) has led to striking progress in photorealistic image synthesis, powered by large-scale generative models such as Stable Diffusion (Rombach et al., 2022), DALL·E (Ramesh et al., 2021), Midjourney (Midjourney, Inc., 2025), and Flux (Labs, 2024). These models can generate high-quality images and are being widely adopted in design, education, media, and science. However, despite visual realism, many AIGC-generated images exhibit subtle but significant *semantic-level anomalies*, such as **logical contradictions**, **physical implausibilities**, or **commonsense violations**, that compromise their authenticity. These inconsistencies highlight the need for structured semantic anomaly detection in AIGC images, which not only facilitates a deeper analysis of such images but also lays the groundwork for explainable deepfake detection and evaluating generative models' reasoning capabilities regarding commonsense knowledge. Furthermore, this approach holds potential for enhancing the semantic coherence of image

---

*Work done during internship at MSRA.
†Corresponding Authors: Yao Zhao(yzhao@bjtu.edu.cn) and Yan Lu(yanlu@microsoft.com)

generation models. Our proposed task directly supports these goals by identifying content-aware violations that compromise semantic authenticity.

As illustrated in Fig. 1(a), such semantic anomalies include hybrid semantics (e.g., mixing sports equipment), violations of physics (e.g., gravity-defying climbers), and anatomical implausibilities. These issues are not captured by traditional low-level forensic cues (Tan et al., 2023; 2024; Durall et al., 2020; Wang et al., 2020; Ojha et al., 2023), yet they critically affect human trust and decision-making when AIGC is used in factual or sensitive domains.

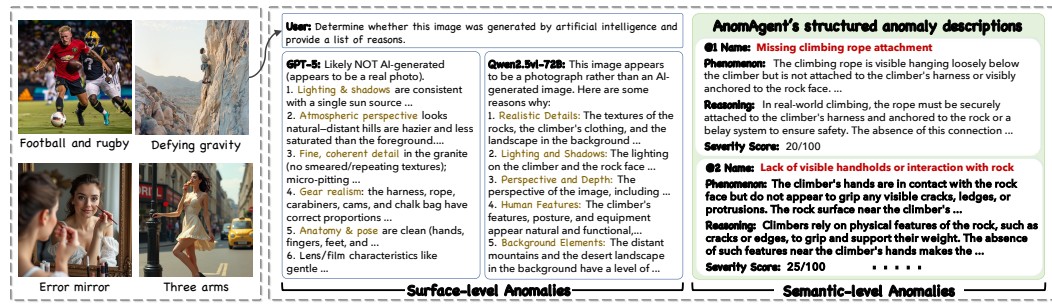

(a) Content-aware semantic anomalies      (b) Detection example using GPT-5, Qwen2.5vl-7B, and AnomAgent

Figure 1: **Semantic anomaly detection in AIGC-generated images.** **(a)** Illustration of high-level semantic anomalies that are context-dependent and subtle, such as inconsistent physics, anatomy, and reflections—challenges that go beyond surface-level visual artifacts. **(b)** Comparison of detection performance between general-purpose vision-language models (e.g., GPT-5, Qwen2.5vl-72B) and the proposed **AnomAgent**. While the former focus on surface-level cues such as lighting and textures, AnomAgent identifies fine-grained semantic inconsistencies and provides structured, explainable outputs with severity ratings.

This work introduces the task of **semantic visual anomaly detection and reasoning** for AIGC image, which seeks to identify and explain *semantic-level anomalies* present in synthetic images. These anomalies pertain specifically to violations of commonsense knowledge, physical plausibility, and logical coherence. Formally defined, given an AIGC-generated image as input, the system is required to produce a set of structured anomaly descriptions comprising four components: ***Name***, ***Phenomenon***, ***Reasoning***, and a corresponding ***Severity Score***, capturing *what* is wrong, *why* it is wrong, and *how severe* it is, as illustrated in Fig. 1(b). Specifically, *Name* provides a concise summary of the anomaly, while *Phenomenon* offers a detailed description at the semantic level. *Reasoning* explains the underlying causes of the anomaly. Finally, *Severity Score* quantifies the anomaly by assigning a score that reflects its authenticity. This formulation underscores not only the importance of detecting anomalies but also providing explanations alongside detailed semantic evaluations.

Several studies (Wen et al., 2025; Gao et al., 2025; Zhang et al., 2025; Zhou et al., 2025) have attempted to extract forgery-related evidence using vision-language models (VLMs) (Bai et al., 2025) from images to support classification outcomes. However, their performance on anomaly detection frequently relies on surface-level irregularities, such as global lighting conditions or shadow patterns, subtle statistical artifacts in texture that are typically imperceptible to human observers (see Fig. 1(b), Left). In contrast, our task focuses on *content-level semantic anomalies*, implausible object interactions, physical violations, or commonsense errors, that are directly visible to humans and therefore more aligned with human judgment (refer to Fig. 1(b), Right). Furthermore, instead of shallow descriptions, the defined *structured anomaly representation* makes anomaly analysis interpretable and accessible, moving beyond detection to structured reasoning.

To support this task, we build **AnomReason**, the first large-scale benchmark for content-aware semantic anomaly detection in AIGC images. AnomReason consists of diverse synthetic scenes annotated with structured semantic anomalies. Each anomaly entry describes what is wrong, why it is wrong, and how severe the inconsistency is capturing errors in object composition, spatial arrangement, interaction logic, or physical constraints. This benchmark is enabled by **AnomAgent**, a modular multi-agent framework that decomposes anomaly reasoning into object perception, attribute analysis, relational reasoning, and anomaly synthesis. To ensure the reliability of annotations

produced by AnomAgent, we incorporate a lightweight **human-in-the-loop verification** stage. This hybrid pipeline balances scale and accuracy by filtering and refining automatically generated results. Compared with purely manual annotation or fully automated generation, our multi-agent plus human verification strategy achieves both interpretability and scalability, allowing AnomReason to provide high-quality structured annotations at unprecedented scale. Furthermore, we propose novel anomaly semantic detection metrics based on Average Precision (AP) and F1-score, referred to as SemAP and SemF1, respectively, to facilitate the evaluation of anomaly semantic detection performance.

By shifting the focus from surface-level artifacts to content-level reasoning, our framework opens new research directions in semantic authenticity, explainable forensics, and commonsense reasoning for generative media. Our proposed benchmark and system, AnomReason and AnomAgent, enable several critical applications: (i) training semantic anomaly detector to gain deeper understanding AIGC images, (ii) developing *explainable deepfake detectors* that not only classify but also provide semantic-level justifications for their predictions, and (iii) conducting *AIGC semantic reasonableness assessment* to evaluate the logical coherence of image generation models. These applications strengthen content authenticity auditing and help guide future AIGC model development.

## 2 ANOMAGENT FRAMEWORK

Detecting semantic anomalies in AI-generated images requires not only visual recognition but also reasoning about commonsense knowledge, physical feasibility, and multi-object interactions. To address this challenge, we propose **AnomAgent**, a modular multi-agent framework designed to emulate human perception and reasoning through structured processes.

As shown in Fig. 2, AnomAgent decomposes the anomaly detection process into three stages: entity parsing, anomaly mining, and structured output generation. Each stage involves specialized agents that collaborate to produce interpretable, high-precision semantic anomaly annotations.

Given an input image $I$, AnomAgent outputs a set of structured anomalies:

$$\mathcal{A} = \{(y_i, o_i, r_i, v_i)\}_{i=1}^{m} \tag{1}$$

where $y_i$ is the anomaly name, $o_i$ the described anomaly phenomenon, $r_i$ the reasoning explains why $o$ is considered anomalous, and $v_i \in [0, 100]$ indicates the severity. A score of 0 denotes implausibility, while 100 represents full realism. We believe that asking the VLM to justify a severity rating encourages deeper reasoning: the model must not only detect an anomaly but also argue why it is mild vs. severe, which we have found leads to richer descriptions.

### 2.1 STAGE 1: VISUAL ENTITY PARSING

Semantic inconsistencies are often object-centric. However, objects in AIGC images can be entangled, distorted, or hallucinated, making entity extraction unreliable. The **Object Recognition Agent** (`ObjectPerceiver`) identifies all semantically distinct entities in the image, with emphasis on human-related objects.

To reduce false negatives, the detection is repeated $T$ times with varying prompts and merged:

$$\mathcal{O}^{(t)} = \texttt{ObjectPerceiver}(I), \quad t = 1, \ldots, T; \quad \mathcal{O} = \bigcup_{t=1}^{T} \mathcal{O}^{(t)} \tag{2}$$

Each object $o_i \in \mathcal{O}$ is represented by an object name and a detailed description.

### 2.2 STAGE 2: MULTI-PERSPECTIVE ANOMALY MINING

Anomalies in AIGC images may arise from incorrect attributes or implausible inter-object interactions. In this stage, we iterate over each object $o_i \in \mathcal{O}$ and perform two complementary forms of semantic consistency analysis: intra-object (attribute-based) and inter-object (relation-based).

**Attribute-Level Analysis.** The **Attribute Analysis Agent** (`AttributeAnalyzer`) examines visual attributes of $o_i$ such as shape, material, and functionality. It identifies internal inconsistencies and produces a set of attribute anomaly candidates:

$$\mathcal{C}_{\text{attr}}^{(i)} = \texttt{AttributeAnalyzer}(o_i) \tag{3}$$

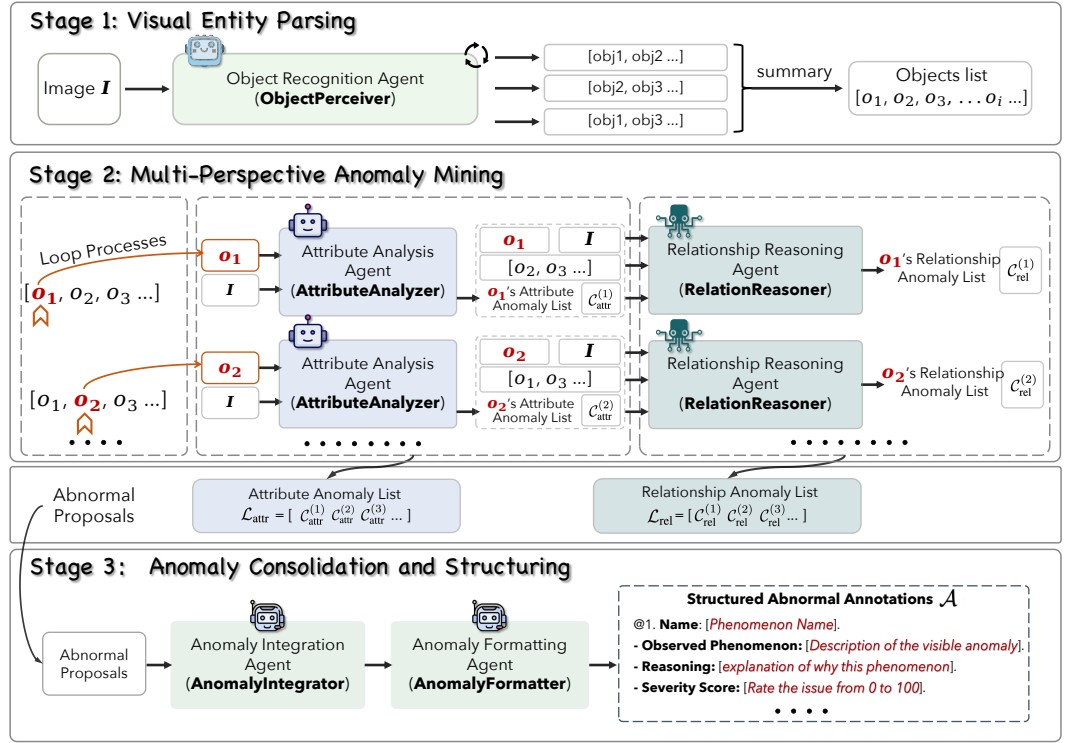

Figure 2: **Overview of the *AnomAgent* pipeline for semantic anomaly annotation.** Stage 1 parses visual entities and yields an object list $\mathcal{O}$. Stage 2 performs multi-perspective anomaly mining, producing attribute candidates $\mathcal{C}_{\text{attr}}$ and relational candidates $\mathcal{C}_{\text{rel}}$, which are scored and pruned to $\mathcal{C}^{+}$. Stage 3 consolidates candidates (merging near-duplicates to $\hat{\mathcal{C}}$) and outputs structured anomalies $\mathcal{A} = \{(y, o, r, v)\}$ (Name, Phenomenon, Reasoning, Severity).

**Relationship-Level Analysis.** The **Relationship Reasoning Agent** (RelationReasoner) evaluates spatial, semantic, and functional interactions between $o_i$ and the rest of the scene, guided by its own attribute anomalies $\mathcal{C}_{\text{attr}}^{(i)}$ as contextual priors:

$$\mathcal{C}_{\text{rel}}^{(i)} = \texttt{RelationReasoner}\left(o_i, \mathcal{O} \setminus \{o_i\}, \mathcal{C}_{\text{attr}}^{(i)}\right) \tag{4}$$

The agent first enumerates pairwise and groupwise relations, then filters semantically implausible ones using both visual context and object-specific inconsistencies. We define the intermediate anomaly lists across all objects as: $\mathcal{L}_{\text{attr}} = \left\{\mathcal{C}_{\text{attr}}^{(i)}\right\}_{i=1}^{|\mathcal{O}|}$, $\mathcal{L}_{\text{rel}} = \left\{\mathcal{C}_{\text{rel}}^{(i)}\right\}_{i=1}^{|\mathcal{O}|}$ The total set of candidate anomalies is: $\mathcal{C} = \bigcup_{i=1}^{|\mathcal{O}|} \left(\mathcal{C}_{\text{attr}}^{(i)} \cup \mathcal{C}_{\text{rel}}^{(i)}\right)$. In addition, to mitigate hallucinations and reduce abnormal omissions, we implement a two-step process in AttributeAnalyzer and RelationReasoner. First, anomalies are comprehensively identified from multiple perspectives; subsequently, these anomalies are verified and structurally outputted.

## 2.3 STAGE 3: ANOMALY CONSOLIDATION AND STRUCTURING

Raw anomaly candidates are noisy, redundant, and linguistically inconsistent. Annotations require clean, standardized outputs. We next consolidate and structure the raw candidate set $\mathcal{C}$ into interpretable anomaly annotations.

**Integration.** The **Anomaly Integration Agent** (AnomalyIntegrator) consolidates overlapping or redundant candidates and removes noise: $\hat{\mathcal{C}} = \texttt{AnomalyIntegrator}(\mathcal{C})$.

**Formatting.** The **Anomaly Formatting Agent** (AnomalyFormatter) maps each anomaly candidate $c \in \hat{\mathcal{C}}$ into a structured four-field annotation: $\mathcal{A} = \{(y_i, o_i, r_i, v_i)\}_{i=1}^{|\hat{\mathcal{C}}|}$. In Fig. 3, we present

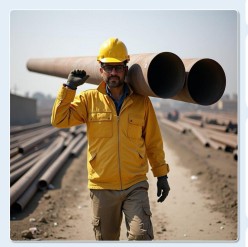

**Name: Implausible pipe placement on shoulder**

**Phenomenon:** Two large cylindrical pipes are resting on the individual's shoulder without visible stabilization or grip. The pipes appear precariously balanced, showing no deformation of the jacket or adjustment in posture to account for their weight.

**Reasoning:** Cylindrical pipes of this size and material would exert significant downward force, causing visible compression of the jacket fabric and requiring the individual to tilt their body or adjust their posture for balance. Additionally, the lack of hand contact or securing mechanism makes it physically implausible for the pipes to remain stable on the shoulder without rolling off.

**Severity Score:** 15

Figure 3: **Example of structured anomalies.** This figure illustrates a detected anomaly where two cylindrical pipes are unrealistically balanced on the individual's shoulder. By structuring the anomaly as {Name, Observed Phenomenon, Reasoning, Severity Score}, the model not only provides a clear description of the anomaly but also offers an interpretable reasoning process, making it easier to understand why this arrangement is physically implausible. The severity score quantifies the degree of implausibility, enhancing the model's ability to observe and explain semantic-level anomalies. This structure allows for transparent and interpretable anomaly detection, improving the detection model's trustworthiness and explainability.

examples of structured anomalies. This structured output enables applications such as quality assessment and deepfake detection.

AnomAgent provides a modular, interpretable, and scalable solution to semantic anomaly detection. By decomposing the reasoning process across multiple agents, it aligns with human judgment and supports practical applications such as quality auditing and explainable detection. Additional details about AnomAgent are provided in Appendix E.

## 3 AnomReason Benchmark

Despite growing interest in semantic anomaly detection, there is no standardized benchmark designed to evaluate *semantic-level* anomalies in AIGC content. We construct **AnomReason**, a large-scale dataset of photorealistic AIGC images annotated with structured semantic anomalies across attribute, relational, and commonsense dimensions.

### 3.1 Data Construction

We collect a diverse set of image-text pairs by crawling approximately 600K user prompts and their corresponding outputs from Midjourney (Midjourney, Inc., 2025). To ensure content realism and diversity, we apply CLIP-based (Radford et al., 2021) filtering on embedding alignment, extracting 109,058 visually realistic samples. A subset of 9,911 images is further manually verified for semantic richness and authenticity.

To enhance generative diversity, we synthesize additional samples using Stable Diffusion 3.5 (AI, 2024) and Flux (Labs, 2024), using the same prompt pool. After automated and manual filtering, we construct a photorealistic dataset comprising 21,539 images: 9,911 from Midjourney, 4,645 from SD3.5, and 6,983 from Flux.

### 3.2 Automatic annotation with AnomAgent

We apply the GPT-4o-based **AnomAgent** framework (Sec. 2) on the full image set, producing 174,872 structured candidate anomalies across 21,539 images. Across the full corpus, the pipeline consumed about 4.17 billion GPT-4o tokens to generate and refine candidate anomalies before HITL screening.[1] Each anomaly contains a textual name, observed phenomenon, commonsense-based reasoning, and a severity rating in $[0, 100]$.

To ensure annotation reliability, we introduce a lightweight *human-in-the-loop (HITL)* quality control stage. Each candidate anomaly $a$ is screened by a trained annotator answering a single-choice

---

[1]We report tokens as aggregated API usage logs, including both prompt and completion.

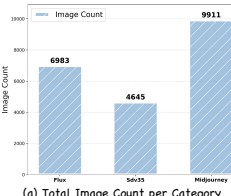 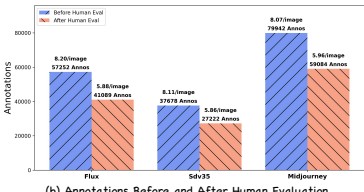 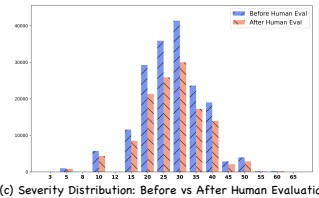

Figure 4: **AnomReason statistics.** (a) Total image count per category: Flux contains 6983 images, Sdv3.5 contains 4645, and Midjourney has the most with 9911 images. (b) Annotations before and after human evaluation: Flux has a reduction from 8.20 to 5.88 annotations per image, Sdv3.5 decreases from 8.11 to 5.86, and Midjourney shows a slight drop from 8.07 to 5.96 annotations per image. (c) Severity distribution before and after human evaluation: It showing a shift towards lower severity values after human evaluation, reflecting the refinement process in the annotation quality.

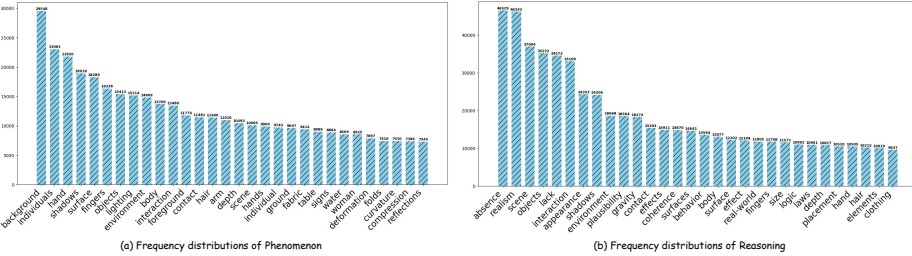

Figure 5: **Top30 noun frequency distributions for the *Phenomenon* and *Reasoning* fields.**

question: *'Is this structured description correct for the given image?'* with three options: AC-CEPT/REJECT/UNSURE.

$$h(a) \in \{1, 0, \perp\} \quad \text{for ACCEPT, REJECT, UNSURE,} \qquad \mathcal{A}_{\text{final}} = \{\, a \in \mathcal{A} : h(a) = 1 \,\}. \quad (5)$$

This low-cost protocol removes implausible hallucinations while preserving structured quality. After HITL filtering, the average valid annotations per image drops from 8 to 5.9, indicating refined semantic focus (Fig. 4).

**Lexical Analysis of Phenomenon and Reasoning fields.** To further characterize the semantic coverage of *AnomReason*, we conduct a corpus-level lexical analysis of the *Phenomenon* and *Reasoning* fields across all verified anomalies. For each field, we extract the head noun of the main semantic unit and compute term frequencies. Figure 5 reports the top-30 most frequent concepts for both fields. The *Phenomenon* distribution shows a balanced mix of object-centric (e.g., 'hand', 'fingers', 'surface'), environment-level ('background', 'shadows'), and interaction-level terms ('contact', 'body interaction'), indicating that anomalies span local appearance, object articulation, and scene-level inconsistencies. The *Reasoning* distribution highlights the dominant semantic principles invoked in explanations, including realism violations ('absence', 'physics', 'plausibility'), object-function mismatches ('interaction', 'affordance'), and contextual logic ('environment', 'coherence'). Together, these distributions suggest that *AnomReason* covers a wide spectrum of semantic error modes, from fine-grained object abnormalities to higher-level physical and commonsense violations, providing rich supervision for training and evaluating reasoning-capable VLM.

**Dataset Split.** We retain **21,533** images for training/testing after removing six duplicates/corrupted items. We adopt a deterministic 50/50 train/test split ($|\mathcal{D}| = 21{,}533$), comprising 10,765 images for training and 10,774 for testing. Each split includes image-level metadata, anomaly counts, severity scores, and intermediate agent outputs to support replicability and ablation studies. We will release all results encompassing intermediate outputs from AnomAgent nodes alongside structured annotations filtered through HIL processing mechanisms.

## 3.3 EVALUATION PROTOCOL

We propose a structure-aware evaluation protocol tailored for semantic-level anomaly reasoning. Each annotation tuple $(y, o, r, v)$ contains a description $o$ and a reasoning field $r$, which are compared

against ground truth using BERTScore (Zhang* et al., 2020). We define three evaluation views: **Phe** (Phenomenon), **Rea** (Reasoning), and **Full** (combined fusion).

We perform one-to-one anomaly matching at the image level based on similarity thresholds $\tau \in \{0.7, 0.8, 0.9\}$, and compute precision/recall curves to derive:

$$\text{SemAP}_v = \frac{1}{|\mathcal{D}|} \sum_{I \in \mathcal{D}} \text{AP}_v(I), \quad \text{SemF1}_v = \frac{1}{|\mathcal{D}|} \sum_{I \in \mathcal{D}} \text{F1}_v(I) \tag{6}$$

Severity scores $v$ can optionally serve as confidence scores in ranking tasks. We adopt the 'distilbert-base-uncased" model for computing BertScore. Comprehensive details are provided in Appendix C.

## 4 EXPERIMENTS

We conduct comprehensive experiments to evaluate the effectiveness of our benchmark, methodology, and task formulation in three progressively structured settings. These experiments are designed to assess both the capabilities of current VLMs and the benefits of targeted fine-tuning for semantic anomaly analysis. First, we evaluate structured semantic anomaly detection and reasoning on the *AnomReason* benchmark, measuring whether VLMs can both identify anomalous phenomena and explain their semantic violations (Sec. 4.1). Second, we extend this reasoning capability to a more applied setting: explainable deepfake detection on *AnomReason-Deepfake*, which requires accurate AI-generated content classification and grounded explanation (Sec. 4.2). Third, we audit modern text-to-image generators by quantifying semantic plausibility and anomaly severity in their outputs using structured reasoning (Sec. 4.3).

Table 1: Comparative performance on the *AnomReason-Test*.

| Model | SemAP$_{\text{Phe}}$ | SemAP$_{\text{Rea}}$ | SemAP$_{\text{Full}}$ | SemF1$_{\text{Phe}}$ | SemF1$_{\text{Rea}}$ | SemF1$_{\text{Full}}$ |
|---|---|---|---|---|---|---|
| LongVA-7B (Zhang et al., 2024) | 0.2579 | 0.2593 | 0.2494 | 0.1641 | 0.1617 | 0.1578 |
| LLaVA-OV-7B (Li et al., 2024) | 0.3280 | 0.2987 | 0.3012 | 0.2044 | 0.1837 | 0.1867 |
| Phi-3.5-Vision (Abdin et al., 2024) | 0.3466 | 0.3040 | 0.3117 | 0.2960 | 0.2572 | 0.2647 |
| MiniCPM-V-2.6 (Hu et al., 2024) | 0.3898 | 0.3501 | 0.3537 | 0.1891 | 0.1698 | 0.1715 |
| InternVL2-8B (Wang et al., 2024b) | 0.3697 | 0.3424 | 0.3424 | 0.3789 | 0.3500 | 0.3507 |
| InternVL2.5-8B (Chen et al., 2024b) | 0.3343 | 0.3070 | 0.3087 | 0.3008 | 0.2733 | 0.2764 |
| InternVL3-8B (Zhu et al., 2025) | 0.4552 | 0.3676 | 0.3927 | 0.1948 | 0.1614 | 0.1703 |
| InternVL3-9B (Zhu et al., 2025) | 0.3871 | 0.3371 | 0.3514 | 0.3953 | 0.3456 | 0.3595 |
| mPLUG-Owl3-7B (Ye et al., 2024) | 0.4026 | 0.3661 | 0.3678 | 0.1247 | 0.1141 | 0.1144 |
| Qwen2-VL-7B (Wang et al., 2024a) | 0.4090 | 0.3564 | 0.3678 | 0.1307 | 0.1208 | 0.1207 |
| InternVL2-26B (Wang et al., 2024b) | 0.4209 | 0.3728 | 0.3865 | 0.4048 | 0.3590 | 0.3722 |
| Qwen2.5-VL-7B (Bai et al., 2025) | 0.4674 | 0.3902 | 0.4155 | 0.4240 | 0.3548 | 0.3775 |
| Qwen2.5-VL-72B (Bai et al., 2025) | 0.4926 | 0.4353 | 0.4568 | 0.4423 | 0.3912 | 0.4104 |
| Gemini-2.5-pro (Google, 2025) | 0.3755 | 0.3127 | 0.3384 | 0.1995 | 0.1674 | 0.1806 |
| GPT-o3 (OpenAI, 2024) | 0.4470 | 0.3762 | 0.4058 | 0.4431 | 0.3724 | 0.4021 |
| GPT-5 (OpenAI, 2025) | 0.3760 | 0.3212 | 0.3469 | 0.4407 | 0.3762 | 0.4065 |
| GPT-4o (Hurst et al., 2024) | 0.4908 | 0.4562 | 0.4727 | **0.5304** | 0.4930 | **0.5109** |
| AnomReasonor-7B | **0.5221** | **0.5130** | **0.5162** | 0.5066 | **0.4977** | 0.5009 |

### 4.1 AIGC SEMANTIC VISUAL ANOMALY DETECTION AND REASONING

The first task evaluates whether VLMs can detect and explain semantic anomalies using structured outputs. The core challenge lies in not only localizing errors but also providing plausible reasoning chains grounded in commonsense or physical priors. We adopt the AnomReason `test` set and use **SemAP** and **SemF1** metrics across phenomenon (Phe), reasoning (Rea), and full (Phe+Rea) views. We fine-tune Qwen2.5-VL-7B via LoRA on the `train` split, resulting in a baseline **AnomReasonor-7B (AR-7B)**. (training details in App. D)

**Results.** Table 1 presents comprehensive results on semantic anomaly detection and reasoning. Overall, most off-the-shelf VLMs struggle with the full task (SemAP$_{\text{Full}}$), with values typically below 0.42, indicating limited semantic understanding in the absence of targeted supervision. Among open-source models, **Qwen2.5-VL-72B** performs best (SemAP$_{\text{Full}}$= 0.4568), followed by InternVL3-8B and InternVL2-26B. However, they still lag behind the top-performing *AnomReasonor-7B*, which achieves a new state-of-the-art **SemAP$_{\text{Full}}$= 0.5162** and the highest reasoning score **SemAP$_{\text{Rea}}$= 0.5130**, demonstrating the effectiveness of fine-tuning on our structured

anomaly supervision. Notably, *AnomReasonor-7B* also surpasses GPT-4o in all **SemAP** metrics. In terms of **SemF1** (alignment quality), GPT-4o retains a slight edge (SemF1$_{Full}$= 0.5109), but *AnomReasonor-7B* is highly competitive (SemF1$_{Full}$= 0.5009), particularly excelling in reasoning (SemF1$_{Rea}$= 0.4977 vs. 0.4930 for GPT-4o). This highlights that our fine-tuned model closes the gap to proprietary systems not only in detection but also in the quality of generated descriptions.

Interestingly, most models show stronger anomaly observation (SemAP$_{Phe}$) than reasoning (SemAP$_{Rea}$), with some (e.g., InternVL3-8B) showing a wide gap (0.4552 vs. 0.3676), suggesting that identifying "something wrong" is easier than articulating "why". In contrast, *AnomReasonor-7B* exhibits the most balanced profile, with reasoning performance nearly matching observation. This reflects the benefit of our structured annotation format and supervision signal, which jointly improve both detection and explanation.

Overall, these results demonstrate that current VLMs exhibit limited semantic anomaly competence in AIGC images, particularly in reasoning. Task-aligned supervision on AnomReason consistently improves both observation and reasoning quality, closing much of the gap to stronger closed-source systems. This highlights the necessity of a semantic-level benchmark and validates the design of our structured supervision.

Table 2: Explainable deepfake detection on *AnomReason-Deepfake*.

| Models | Acc | CSemAP$_{Phe}$ | CSemAP$_{Rea}$ | CSemAP$_{Full}$ | CSemF1$_{Phe}$ | CSemF1$_{Rea}$ | CSemF1$_{Full}$ |
|---|---|---|---|---|---|---|---|
| LongVA-7B (Zhang et al., 2024) | 53.45 | 0.0722 | 0.0714 | 0.0688 | 0.0458 | 0.0448 | 0.0436 |
| LLaVA-OV-7B (Li et al., 2024) | 66.26 | 0.1235 | 0.1124 | 0.1141 | 0.0818 | 0.0738 | 0.0752 |
| Phi-3.5-Vision (Abdin et al., 2024) | 41.33 | 0.0685 | 0.0602 | 0.0616 | 0.0570 | 0.0497 | 0.0511 |
| MiniCPM-V-2.6 (Hu et al., 2024) | 55.05 | 0.0233 | 0.0205 | 0.0209 | 0.0109 | 0.0100 | 0.0100 |
| InternVL2-8B (Wang et al., 2024b) | 59.46 | 0.0739 | 0.0702 | 0.0697 | 0.0720 | 0.0680 | 0.0678 |
| InternVL2.5-8B (Chen et al., 2024b) | 63.61 | 0.1165 | 0.1085 | 0.1089 | 0.0988 | 0.0911 | 0.0919 |
| InternVL3-8B (Zhu et al., 2025) | 62.84 | 0.2949 | 0.2394 | 0.2559 | 0.1240 | 0.1031 | 0.1089 |
| InternVL3-9B (Zhu et al., 2025) | 64.04 | 0.2293 | 0.1987 | 0.2081 | 0.2343 | 0.2041 | 0.2132 |
| mPLUG-Owl3-7B (Ye et al., 2024) | 55.96 | 0.0445 | 0.0411 | 0.0404 | 0.0130 | 0.0120 | 0.0117 |
| Qwen2-VL-7B (Wang et al., 2024a) | 67.21 | 0.1710 | 0.1421 | 0.1496 | 0.0524 | 0.0458 | 0.0467 |
| InternVL2-26B (Wang et al., 2024b) | 54.73 | 0.0194 | 0.0172 | 0.0179 | 0.0180 | 0.0160 | 0.0166 |
| Qwen2.5-VL-7B (Bai et al., 2025) | 65.41 | 0.1295 | 0.1085 | 0.1155 | 0.1124 | 0.0943 | 0.1004 |
| Qwen2.5-VL-72B (Bai et al., 2025) | 77.60 | 0.2626 | 0.2337 | 0.2453 | 0.2427 | 0.2159 | 0.2266 |
| Gemini-2.5-pro (Google, 2025) | 85.65 | 0.2631 | 0.2192 | 0.2382 | 0.1488 | 0.1249 | 0.1351 |
| GPT-o3 (OpenAI, 2024) | 85.60 | 0.3189 | 0.2690 | 0.2898 | 0.3187 | 0.2685 | 0.2895 |
| GPT-5 (OpenAI, 2025) | 75.22 | 0.1790 | 0.1535 | 0.1658 | 0.2114 | 0.1811 | 0.1957 |
| GPT-4o (Hurst et al., 2024) | **87.76** | **0.3750** | 0.3487 | 0.3612 | **0.4054** | 0.3770 | 0.3905 |
| AnomReasonor-7B | 82.61 | 0.3684 | **0.3574** | **0.3613** | 0.4048 | **0.3929** | **0.3972** |

## 4.2 EXPLAINABLE DEEPFAKE DETECTION

We extend our semantic anomaly reasoning framework (Sec. 4.1) to an explainable deepfake detection setting. The goal is twofold: (i) determine whether an image is AI-generated, and (ii) return structured anomaly explanations aligned with Sec. 3.3. To this end, we construct **AnomReason-Deepfake**, where real images sampled from LAION/reLAION-HR (LAION e.V., 2023) are paired with content-based structured descriptions. The task therefore probes both AIGC detection and semantic explanation within a unified benchmark.

**Metrics.** In addition to binary detection accuracy (Acc), we introduce classification-aware variants of our semantic metrics, denoted as $\mathrm{CSemAP}_v$ and $\mathrm{CSemF1}_v$ for $v \in \{\mathrm{Phe, Rea, Full}\}$. Explanations are scored only when detection is correct, and assigned zero otherwise. This ties explanation quality to valid classification, discouraging post-hoc rationalization for mispredictions and promoting joint interpretability.

**Results.** Table 2 presents results across Acc and classification-aware semantics metrics. GPT-4o achieves the best overall detection accuracy (87.76%) and strong CSemF1$_{Full}$(0.3905), establishing a strong proprietary baseline. Notably, our fine-tuned *AnomReasonor-7B* attains competitive accuracy (82.61%) and **surpasses** GPT-4o on CSemAP$_{Rea}$(0.3574 vs. 0.3487) and CSemF1$_{Rea}$(0.3929 vs. 0.3770), highlighting its strength in generating causally grounded explanations. Open-source VLMs show large variance. Larger models like Qwen2.5-VL-72B achieve decent detection (77.60%) and moderate explanation ability (CSemF1$_{Full}$: 0.2266), but remain behind task-aligned models. Some models (e.g., InternVL3-8B) perform reasonably on CSemAP$_{Full}$(0.2559) but poorly on F1, suggesting limited calibration between confidence and semantic consistency. Overall, *AnomReasonor-7B* offers a high-accuracy, interpretable alternative to closed models. These results underscore the util-

ity of *AnomReason-Deepfake* as a testbed for **joint detection and structured explanation**, and support our pipeline's effectiveness for training explainable AIGC detectors.

Compared with general-purpose VLMs, AnomReasonor achieves substantially stronger performance when both classification and explanation are required. While the focus of our work is semantic reasoning rather than low-level forgery cues, the classification-aware metrics demonstrate that semantic inconsistency signals provide complementary benefits to traditional deepfake detection. This suggests that semantic anomaly reasoning forms a useful capability orthogonal to specialized forensic detectors.

Table 3: Semantic Reasonableness of AIGC Image Generators.

| AIGC Model | AnomReasonor | | | AnomAgent | | |
|---|---|---|---|---|---|---|
| | MAI ($\downarrow$) | AF($\downarrow$) | CAP($\downarrow$) | MAI ($\downarrow$) | AF($\downarrow$) | CAP($\downarrow$) |
| Sana 1.5 (Xie et al., 2025) | 4.55 | 6.29 | 28.62 | 6.66 | 9.09 | 60.49 |
| SDXL Lightning (Lin et al., 2024) | 4.40 | 6.06 | 26.66 | 6.47 | 8.84 | 57.21 |
| Sana Sprint 1.6B (Chen et al., 2025a) | 4.32 | 6.09 | 26.31 | 6.42 | 8.78 | 56.35 |
| Qwen-lmage (Wu et al., 2025a) | 4.45 | 6.20 | 27.59 | 6.37 | 8.72 | 55.61 |
| HiDream-1-Fast (Cai et al., 2025) | 4.44 | 6.19 | 27.48 | 6.32 | 8.81 | 55.68 |
| SDv3.5 Large (Esser et al., 2024) | 4.60 | 6.44 | 29.62 | 6.23 | 8.54 | 53.19 |
| Janus Pro 7B (Chen et al., 2025b) | 4.28 | 6.75 | 28.89 | 6.49 | 8.69 | 56.39 |
| Janus Pro 1B (Chen et al., 2025b) | 4.19 | 6.43 | 26.94 | 6.40 | 8.41 | 53.83 |
| FLUX.1 [dev] (Wu et al., 2025a) | 4.36 | 6.08 | 26.51 | 6.22 | 8.67 | 53.92 |
| BRIA-3.2  (Bria AI, 2025) | 4.38 | 6.14 | 26.89 | 6.11 | 8.60 | 52.61 |
| SDv3.5 Large Turbo (Esser et al., 2024) | 4.40 | 6.12 | 26.93 | 6.10 | 8.44 | 51.48 |
| Lumina lmage V2 (Qin et al., 2025) | 4.30 | 6.05 | 26.01 | 6.07 | **8.40** | 51.03 |
| HiDream-1-Dev (Cai et al., 2025) | 4.42 | 6.15 | 27.18 | **6.00** | 8.44 | **50.62** |
| OmniGen V2 (Wu et al., 2025b) | 4.21 | **5.89** | 24.80 | 6.11 | 8.49 | 51.86 |
| HunyuanImage-2.1 (Team, 2025b) | **4.10** | 5.95 | **24.40** | 6.14 | 8.55 | 52.51 |

### 4.3 AIGC SEMANTIC REASONABLENESS ASSESSMENT

Beyond instance-level evaluation (Sec. 4.1), we assess the semantic reasonableness of text-to-image generators. Perceptual metrics overlook commonsense, physics, and interaction plausibility, whereas structured outputs (Name/Phenomenon/Reasoning/Severity Score) enable semantics-aware auditing at scale.

We curate 246 prompts from Midjourney public galleries that correspond to real-photo style content. We first deduplicate/cluster candidate prompts via CLIP embeddings, then filter to photo-realistic style using the Qwen3-30B (Team, 2025a). This yields diverse prompts covering multi-object interactions, human articulation, and physical dynamics. Each generated image is evaluated independently by two assessors: (i) *AnomReasonor-7B* (fine-tuned in Sec. 4.1) and (ii) *AnomAgent*. Both produce anomalies with an Severity Score $s \in [0, 100]$.

**Semantic Metrics.** We define three complementary metrics to quantify the semantic plausibility of an image $I$: (i) **MeanAnomalyImplausibility (MAI):** $\text{MAI}(I) = \sum_{\hat{a} \in \widehat{\mathcal{A}}(I)} \frac{100 - s(\hat{a})}{100}$ (ii) **AnomalyFrequency (AF):** Total number of anomalies $\hat{a}$ identified in $I$. (iii) **CumulativeAnomalyPenalty (CAP):** $\text{CAP}(I) = \text{MAI}(I) \times \text{AF}(I)$ The MAI captures the aggregated implausibility of all detected anomalies. CAP reflects both the severity and frequency of semantic violations. All scores are designed such that **lower is better**, with zero indicating flawless realism.

**Results.** Table 3 reveals clear differences in semantic plausibility across AIGC models. HunyuanImage-2.1 and OmniGen V2 achieve the lowest CAP scores under AnomReasonor, indicating fewer and less severe semantic anomalies, particularly in commonsense and physical interactions. In contrast, models like Sana 1.5 and SDXL Lightning exhibit higher anomaly frequency and implausibility, suggesting challenges in compositional reasoning and realism. Interestingly, we observe distinct failure modes: some models (e.g., Janus Pro 7B) exhibit higher AF but lower MAI, implying frequent yet subtle errors, while others (e.g., SDv3.5 Large) show fewer but more severe implausibilities. These insights go beyond perceptual quality, revealing gaps in physical logic and social commonsense. While both AnomReasonor and AnomAgent show consistent relative rankings, AnomAgent tends to produce higher anomaly counts across the board, reflecting its greater sensitivity and fully automated nature. Despite this, their strong rank-order alignment supports AnomAgent's robustness and suitability for scalable zero-shot auditing. Notably, AnomReasonor benefits from supervised training on human-curated explanations, whereas AnomAgent

operates without any human intervention—highlighting the promise of fully automatic, semantics-aware evaluation pipelines for future alignment assessments.

The cross-model assessment reveals large semantic variability among state-of-the-art generative models, indicating that high perceptual quality does not guarantee semantic plausibility. Anom-Reasonor provides a scalable, model-agnostic diagnostic that exposes systematic weaknesses in physical, contextual, and interaction logic, offering a foundation for evaluating and improving next-generation AIGC systems.

## 5   RELATED WORK

**AIGC visual anomaly detection and reasoning:**   With the proliferation of AI-generated content (AIGC), recent research (Liu et al., 2024; Huang et al., 2025a; Wen et al., 2025; Kang et al., 2025; Gao et al., 2025; Huang et al., 2025b; Zhang et al., 2025; Zhou et al., 2025; Tan et al., 2025b; Guo et al., 2025; Chen et al., 2024a) has shifted beyond low-level forensics to explore semantic-level anomalies in AIGC images. Some methods leverage VLM interpretability to highlight suspicious regions (Liu et al., 2024; Huang et al., 2025a; Kang et al., 2025), while others combine manual inspection with LLM-based post-processing (Li et al., 2025; Tan et al., 2025a). Prompt-engineering and commonsense-injection techniques have also been used to elicit finer-grained descriptions (Zhang et al., 2025; Gao et al., 2025; Wen et al., 2025; Zhou et al., 2025).

However, existing benchmarks such as FakeClue and Ivy-Fake focus on authenticity classification or artifact explanation. They provide coarse labels or clues for real/fake discrimination, but lack structured quadruple annotations that capture commonsense, physical and relational inconsistencies. Consequently, models trained on them struggle to perform detailed reasoning or severity assessment. Our work differs by modelling anomalies at the object–attribute–relation level and offering interpretable explanations, thus enabling downstream tasks like semantic reasonableness auditing. We also adopt a multi-agent annotation framework with human verification, which yields more consistent and scalable annotations than directly prompting a monolithic LLM.

**AIGC Image Semantic Reasonableness Assessment:**   Assessing the *semantic reasonableness* of AI-generated images (AIGC-ISRA) involves determining whether the visual content aligns with real-world commonsense, object plausibility, and coherent inter-object relationships. While current AIGC image quality assessment efforts (Peng et al., 2024; Liu et al., 2023; Lu et al., 2023; Yang et al., 2024; Yu et al., 2024) focuses primarily on image–prompt alignment and perceptual quality, but overlook semantic plausibility. For example, (Peng et al., 2024) propose CLIP-based metrics to assess prompt consistency, yet they fail to capture scene-level semantic inconsistences. (Liu et al., 2023) explore chain-of-thought evaluation for NLG tasks, offering improved human alignment but lacking grounding and localized reasoning for visual content. Our work addresses this gap via a structured and content-aware evaluation framework that builds upon a multi-agent annotator (*AnomAgent*) to detect and explain attribute violations, spatial logic failures, and inter-object contradictions. This enables interpretable and fine-grained assessment of semantic plausibility, bridging a key limitation of existing AIGC-IQA and VLM-based approaches.

## 6   CONCLUSION

We introduce the task of *semantic anomaly detection and reasoning* in AI-generated image. To enable this task, we built *AnomReason*, a benchmark annotated via a multi-agent framework (*AnomAgent*) and human verification, providing structured quadruples (Name, Phenomenon, Reasoning, Severity). Unlike existing authenticity datasets, AnomReason targets commonsense, physical and relational violations and includes severity scores, enabling finer-grained reasoning beyond real/fake classification. We propose semantics-aware metrics and showed that off-the-shelf vision-language models struggle on this task; a model fine-tuned on AnomReason (AnomReasonor-7B) outperformed open-source baselines and approached proprietary systems. We also demonstrated applications in explainable deepfake detection and generator assessment. Current limitations include the dataset's moderate scale and focus on images; future work will extend to videos and refine annotation quality. We will release code, data and models to foster research into semantic plausibility and more trustworthy multimodal systems.

ACKNOWLEDGMENTS

This work was supported in part by the National Natural Science Foundation of China (No.U24B20179, No.62506030, Grant U23B2022, No.62336001), Beijing Natural Science Foundation (No. 4264127), China Postdoctoral Science Foundation(Grant No. 2025M781453) and Shenzhen R&D Program(Grant JCYJ20250604181211016, SYSPG20241211174032004).

ETHICS STATEMENT

This work adheres to ethical principles surrounding data usage, model training, and large-scale evaluation. All datasets used in this study are either synthetic, automatically generated from text prompts (e.g., via Midjourney), or released under open-source licenses. No personally identifiable information (PII), biometric data, or user-generated content is used. All visual data used in training or evaluation is publicly available or synthesized, and filtered to avoid unsafe or offensive content.

Human annotations used in the *AnomReason* dataset were collected through internal quality-controlled pipelines, focusing solely on semantic inconsistencies in AI-generated content. These annotations are descriptive, non-judgmental, and pertain to visual plausibility, physics, and common-sense violations rather than identity or sensitive topics. Annotators were not exposed to real-world sensitive content. Additionally, we introduced the *AnomAgent*, an automated system that detects and explains semantic anomalies in images without human involvement, relying on large vision-language models (e.g., GPT-4o, Qwen2.5-VL). While the *AnomReasonor* model was fine-tuned on filtered, verified annotations, *AnomAgent* is entirely zero-shot, demonstrating generalization without human bias injection during training.

This work also uses LLMs (e.g., GPT-4o) for data annotation and linguistic refinement. For annotation, LLMs are used under structured prompts that enforce semantic rigor and reduce hallucination. For writing, they were used only for grammar polishing, as stated in Section A, and did not contribute conceptually.

The proposed methods aim to improve the semantic transparency and quality of AIGC systems. While our anomaly detection framework could assist in automated auditing or filtering pipelines, we explicitly caution against its use in high-stakes decision-making or content moderation without human oversight. We strongly discourage deployment for surveillance or enforcement purposes, and advocate for responsible use aligned with human-centered values, fairness, and transparency.

REPRODUCIBILITY STATEMENT

In accordance with the ICLR Author Guide, we provide a detailed reproducibility statement for this work.

**Dataset Availability.** The AnomReason benchmark was constructed from 21,539 AI-generated images, sourced from Midjourney, Stable Diffusion 3.5, and Flux. All images are synthetic and filtered for realism using CLIP-based selection and manual checks. We will release the complete dataset, including raw images, structured anomaly annotations, severity scores, and metadata.

**Annotation Pipeline.** Annotations were generated using the AnomAgent multi-agent system with GPT-4o, refined through human-in-the-loop (HITL) verification. The pipeline produced 174,872 candidate anomalies, reduced to an average of 5.9 verified anomalies per image. Intermediate outputs from each agent stage will also be released to support ablation studies.

**Model Training.** We fine-tuned Qwen2.5-VL-7B using LoRA adapters inserted at every fourth transformer layer. The visual encoder was frozen. Training used 4 NVIDIA A6000 GPUs, batch size of 4, and gradient accumulation of 8, for one epoch over the training split. LoRA rank was 8 with scaling factor $\alpha = 16$ and dropout 0.5. All hyperparameters and training scripts will be provided.

**Evaluation Protocol.** We designed semantic evaluation metrics, SemAP and SemF1, based on BERTScore similarity between predicted and ground-truth anomaly fields. We tested thresholds $\tau \in \{0.7, 0.8, 0.9\}$ and reported averages. We will release the full evaluation toolkit, including BERTScore settings, matching rules, and baseline implementations.

**Released Artifacts.** We will publicly release:

- The AnomReason dataset and splits.
- All anomaly annotations with HITL verification.
- The AnomReasonor-7B fine-tuned model weights.
- Code for training, evaluation.

**Limitations to Reproducibility.** Some annotation stages depend on external APIs (e.g., GPT-4o), which may introduce variability due to model updates. To mitigate this, we release all intermediate outputs used in the dataset construction. The HITL step, while low-cost, involves subjective human judgment and may not be identically reproducible.

**Summary.** All code, data, models, and evaluation protocols will be released under an open-source license to maximize reproducibility and transparency.

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

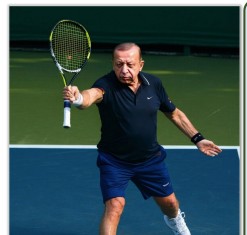

**@1 Name: Foreground and Background Lighting Mismatch**

**Phenomenon:** The lighting on the tennis court foreground is bright and even, while the background wall appears dimmer and less illuminated. This creates a noticeable contrast that disrupts the scene's coherence.

**Reasoning:** In real-world photography, foreground and background elements should share similar lighting conditions to maintain visual consistency. The stark difference in illumination suggests a mismatch between the generated elements.

**Severity Score:** 20

**@2 Name: Foreground and Background Depth Mismatch**

**Phenomenon:** The tennis court foreground appears sharply defined and in focus, while the background wall lacks depth and detail. The transition between the two areas is abrupt, creating a sense of separation rather than integration.

**Reasoning:** Real-world scenes typically exhibit gradual depth changes, with objects in the foreground appearing sharper and more defined, while those in the background become progressively softer and less detailed. The abrupt transition here disrupts spatial coherence.

**Severity Score:** 20

**@3 Name: Foreground and Background Color Mismatch**

**Phenomenon:** The tennis court foreground has a uniform blue color, while the background wall appears green. This color inconsistency is particularly noticeable around the edges where the two areas meet.

**Reasoning:** Real-world scenes typically exhibit consistent color gradients across adjacent surfaces. The abrupt change in color at the edge of the court and wall suggests a mismatch between the generated elements.

**Severity Score:** 20

**@4 Name: Right Arm and Wrist Movement**

**Phenomenon:** The right arm is extended outward, but the wrist and fingers are unnaturally stiff and lack visible motion or tension. The wristband is also misaligned, overlapping the wrist and extending awkwardly over the forearm.

**Reasoning:** In real-life tennis, players use their arms and wrists dynamically during swings, with visible tension and motion. The stiffness and misalignment of the right arm and wristband suggest an artificial overlay rather than a natural interaction.

**Severity Score:** 25

**@5 Name: Tennis Racket Grip and Hand Positioning**

**Phenomenon:** The tennis racket grip appears overly tight, with the hand positioned unnaturally close to the handle. The wristband on the left arm is also misaligned, overlapping the wrist and extending awkwardly over the forearm.

**Reasoning:** In real-life tennis, players grip the racket loosely for better control and flexibility. The tight grip and misaligned wristband suggest an artificial overlay rather than a natural interaction between the hand and racket.

**Severity Score:** 25

**@6 Name: Forearm and Wrist Alignment**

**Phenomenon:** Human forearms and wrists naturally curve slightly due to muscle tension and bone structure. The rigid posture disrupts the anatomical plausibility of the arm's movement.

**Reasoning:** In real-life tennis, players grip the racket loosely for better control and flexibility. The tight grip and misaligned wristband suggest an artificial overlay rather than a natural interaction between the hand and racket.

**Severity Score:** 30

Figure 6: **Structured visual anomalies in a tennis scene.** AnomReasonor-7B identifies both surface-level inconsistencies (e.g., lighting and color mismatch) and deeper semantic-level anomalies, such as biomechanically implausible wrist articulation and unnatural hand–racket interaction. Each anomaly is described with a structured triplet: Name, Phenomenon, Reasoning.

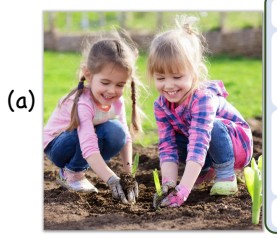

(a)

**Name: Plant growth anomaly**

**Phenomenon:** The seedlings being planted appear unusually tall and slender compared to typical young plants. They lack natural curvature or branching, and their leaves are disproportionately large relative to the stems.

**Reasoning:** Young plants typically grow in a more compact and robust manner, with leaves developing gradually as they mature. The exaggerated height and leaf size suggest an artificial exaggeration rather than a realistic representation of plant growth.

**Severity Score:** 20

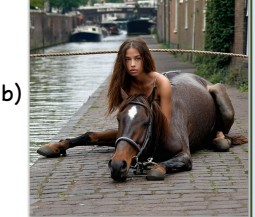

(b)

**Name: Anatomical inconsistency in the horse's body posture**

**Phenomenon:** The horse appears to be lying down, but its front legs are unnaturally positioned, with one leg bent at an angle that defies natural anatomy and weight distribution. Additionally, the horse's head is tilted forward, which does not align with typical resting behavior for a horse.

**Reasoning:** Horses naturally lie down by bending their front legs at the knees and placing them under their bodies. The observed positioning suggests either a mechanical error or an attempt to create a more dynamic pose, which is inconsistent with the horse's anatomy and behavior.

**Severity Score:** 25

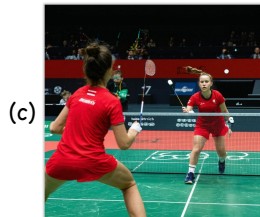

(c)

**Name: Inconsistent ball trajectory and motion blur**

**Phenomenon:** The shuttlecock appears to be in mid-air but lacks realistic motion blur, suggesting it was not recently hit. Additionally, the shuttlecock's position relative to the players does not align with typical badminton gameplay dynamics.

**Reasoning:** Shuttlecocks typically exhibit significant motion blur when in flight due to their fast speed and aerodynamic properties. Furthermore, the shuttlecock's position suggests it was hit by one player but not yet returned, which is inconsistent with the active gameplay depicted.

**Severity Score:** 25

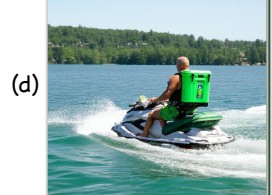

(d)

**Name: Water spray directionality**

**Phenomenon:** The water spray created by the jet ski is directed primarily forward and slightly upward, rather than creating a more dynamic arc or fan-like pattern. The spray appears less dispersed and lacks the expected turbulence and motion associated with high-speed water.

**Reasoning:** High-speed jet skis typically create a more pronounced and dynamic spray pattern due to the force exerted on the water. The observed spray direction and dispersion are inconsistent with real-world physics, making it appear artificial and unrealistic.

**Severity Score:** 20

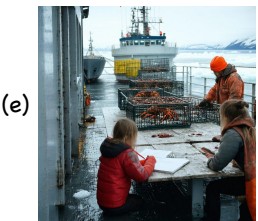

(e)

**Name: Person in orange jacket**

**Phenomenon:** The person wearing the orange jacket appears to be partially obscured by the table, yet their arm and hand remain visible and in focus. This creates an optical illusion where the person's body seems to pass through the table.

**Reasoning:** Objects cannot physically pass through solid surfaces like tables. The visibility of the person's arm and hand defies the laws of physics and spatial logic, making the scene appear unrealistic.

**Severity Score:** 30

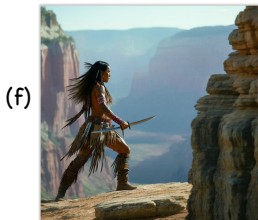

(f)

**Name: Unrealistic Hair Dynamics**

**Phenomenon:** The character's hair appears unnaturally stiff and straight, with no visible motion or interaction with the wind or surrounding environment. The strands do not bend, sway, or align with the character's posture or the canyon's airflow.

**Reasoning:** Human hair responds dynamically to environmental factors like wind, gravity, and movement. The absence of these effects makes the hair appear artificial and out of place within the scene.

**Severity Score:** 20

Figure 7: **Generalization across diverse semantic anomaly types.** AnomReasonor-7B detects inconsistencies in plant growth patterns (a), horse anatomy (b), shuttlecock dynamics (c), water spray physics (d), occlusion logic (e), and hair movement (f). These outputs demonstrate the model's capability to reason over visual semantics beyond surface-level cues.

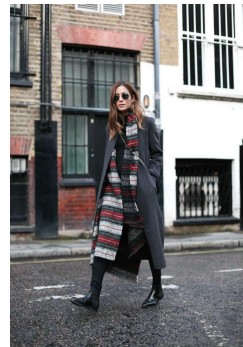

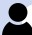 Summarize whether this image is Generated by Artificial Intelligence, please only return yes or no.

No, this image is a real photograph. 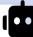

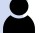 Output your reasoning. If AI-generated, list semantic anomalies: Name, Phenomenon, Reasoning, Severity.

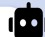

### Description of the Image:

The image depicts an individual walking on a street in an urban environment. The person is dressed in a stylish and layered outfit suitable for cooler weather. Key elements of the attire include:

1. **Outerwear**: The individual is wearing a long, dark gray coat that reaches mid-calf length. The coat appears to be made of a woolen material, suitable for cold weather.

2. **Scarf**: A colorful scarf with a striped pattern in red, white, and green is draped around the neck. The scarf adds a vibrant contrast to the otherwise neutral tones of the outfit.

3. **Inner Layer**: Beneath the coat, there is a black top visible at the neckline, suggesting a layered look.

4. **Bottoms**: The individual is wearing black leggings or tights, which are partially visible under the coat.

5. **Footwear**: Black ankle boots with a slight heel are worn, completing the outfit.

### Background Elements:

- The setting is an urban street with a mix of architectural styles.

- The buildings in the background have a traditional design, featuring brick facades and large windows with metal grilles. One building has a red brick section, while another has a white facade with black grilles.

- The street surface is asphalt, and there are visible road markings, including a yellow line indicating a no-parking zone.

- The overall lighting suggests it might be daytime, with natural light illuminating the scene.

### Analysis and Reasoning:

This image captures a fashion-oriented individual in an urban setting. The combination of the layered clothing, vibrant scarf, and stylish footwear suggests an emphasis on fashion and personal style. The urban backdrop provides context, indicating the photo was taken in a city environment. The individual's pose and attire suggest they may be posing for a fashion-related purpose, such as a street-style photo shoot or a casual outing.

### Final Summary:

The image shows a person walking on a street in an urban area, dressed in a stylish outfit consisting of a long gray coat, a colorful striped scarf, black leggings, and black ankle boots. The background features traditional brick buildings with metal grilles, indicating a city setting. The overall composition suggests a focus on fashion and urban aesthetics.

Figure 8: **Real-image evaluation example.** Under the two-round protocol, the model predicts the image is *real* (Round 1), and in Round 2 produces only a neutral scene description without anomaly annotations. This demonstrates correct suppression of anomaly reasoning on a commercial-style photograph.

## A   THE USE OF LARGE LANGUAGE MODELS (LLMS)

In accordance with ICLR 2026 Author Guide, we disclose the use of large language models (LLMs) at two stages of this work:

(1) **Language Refinement**: OpenAI's GPT-4o is used solely for grammar and fluency enhancement during manuscript preparation. It do not contribute to the conceptual development, experimental

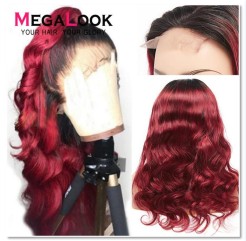
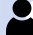
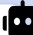
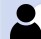

Summarize whether this image is Generated by Artificial Intelligence, please only return yes or no.

No, this image is a real photograph.

Output your reasoning. If AI-generated, list semantic anomalies: Name, Phenomenon, Reasoning, Severity.

**Description of the Image:**

The image appears to be an advertisement for hair products, specifically wigs or hair extensions. It features three distinct views of a wig with a gradient color scheme and wavy texture.

1. **Left Side (Top View):**

  - The top view shows a wig placed on a mannequin head. The wig has a lace front, which is visible at the top of the image.

  - The hair color transitions from dark brown at the roots to vibrant red at the tips, creating a gradient effect.

  - The hair is styled in loose waves, giving it a voluminous and textured appearance.

  - The background includes a tripod stand, suggesting that the image was professionally taken.

2. **Right Side (Top View):**

  - This view shows another wig placed on a mannequin head, similar to the left side but with a slightly different angle.

  - The hair color and style are consistent with the left side, featuring a gradient from dark brown to red and loose waves.

  - The lace front is also visible, indicating that this is likely a lace-front wig designed for natural hair integration.

3. **Bottom Right (Side View):**

  - This view provides a side perspective of the same wig as seen in the previous two images.

  - The hair is styled in loose waves, and the gradient color transition is clearly visible.

  - The texture and volume of the hair are emphasized, showcasing its fullness and movement.

4. **Text Overlay:**

  - At the top left corner of the image, there is text that reads \"MEGALOOK\" in bold, colorful letters. Below it, smaller text states \"YOUR HAIR YOUR GLORY.\"

  - The text suggests that the image is promoting a brand or product related to hair styling or extensions.

### Reasoning:

- The presence of mannequin heads and professional lighting indicates that this is a promotional image intended for advertising purposes.

- The consistent styling and color gradient across all three views suggest that these are examples of the same product being showcased from different angles.

- The text overlay reinforces the commercial intent of the image, highlighting the brand name and slogan.

### Conclusion:

This image is a professional advertisement for a hair product, likely a wig or hair extension, featuring a gradient color scheme and loose wave styling. The text overlay identifies the brand as \"MEGALOOK,\" emphasizing the product's focus on enhancing hair appearance.

Figure 9: **Real-image evaluation example.** The model again answers *real* in Round 1 and outputs a coherent, human-like description of the photographed scene in Round 2. No anomaly hallucinations occur, reflecting strong generalization to natural street photography.

design, data analysis, or substantive content generation. All technical ideas and results are the original work of the authors.

(2) **Data Annotation Assistance**: LLMs, including GPT-4o and Qwen, are employed as annotation agents to assist in generating structured anomaly descriptions for AI-generated images, following

a constrained prompt-response schema designed by the authors. These models are used to generate candidate annotations or supervise anomaly detection model training (e.g., AnomReasonor, AnomAgent). All prompt designs, schema definitions, quality control procedures, and usage decisions were made by the authors, who maintained full control and responsibility over data curation.

The use of LLMs is strictly bounded to assistive roles; they are not involved in authorship, nor in the scientific reasoning or conclusions of this work.

## B  DETECTION RESULTS OF ANOMREASONOR

### B.1  QUALITATIVE ANALYSIS

We present qualitative results from *AnomReasonor-7B*, trained on our proposed *AnomReason* dataset, to analyze its ability to detect and explain semantic-level visual anomalies in AI-generated images. Unlike surface-level cues—such as pixel artifacts, texture mismatches, or inconsistent lighting—semantic-level anomalies reflect inconsistencies with real-world physical laws, common-sense reasoning, spatial logic, and anatomical plausibility.

Figures 6–7 highlight how structured reasoning outputs—composed of `Name`, `Phenomenon`, `Reasoning`, and `Severity Score`—enable fine-grained interpretation of visual errors that would be challenging to localize using traditional classification-based or pixel-level approaches.

Figure 6 presents a synthesized tennis court scene containing a variety of both surface- and semantic-level anomalies. The model detects lighting, depth, and color mismatches between foreground and background, indicating surface inconsistencies. More critically, it identifies human-centric semantic anomalies including biomechanically implausible wrist articulation, unnatural racket grip tension, and anatomically misaligned arm–forearm joints. These violations of physical and anatomical realism are explicitly explained, demonstrating the model's understanding of dynamic human interaction and contextual coherence.

Figure 7 demonstrates the model's ability to generalize across diverse contexts and anomaly types. Examples include:

> **(a)** exaggerated plant growth violating botanical development norms,
>
> **(b)** a horse in an anatomically impossible lying posture,
>
> **(c)** a shuttlecock trajectory inconsistent with typical gameplay physics,
>
> **(d)** jet ski spray patterns that defy real-world water dynamics,
>
> **(e)** human limbs passing through occluding surfaces,
>
> **(f)** static hair ignoring environmental effects such as gravity or wind.

These structured explanations go beyond low-level discrepancies, offering human-aligned insight into the visual implausibility of each scene.

These results showcase the benefit of *semantic-level anomaly detection* for building interpretable and trustworthy AIGC assessment systems. By explicitly modeling and articulating inconsistencies in visual semantics—rather than relying on post-hoc explanations or pixel cues—*AnomReasonor-7B* provides rich, localized, and human-aligned insights into failure modes in AI-generated images.

### B.2  RESULTS ON REAL IMAGES

To assess whether the model overfits to AIGC-specific artifacts or can generalize to natural images, we additionally evaluate AnomReasonor on a set of real-world photographs (examples in Fig. 8 and Fig. 9). We adopt the two-round protocol described in App. D.2: (1) the model first answers a binary question (*real* vs. *AI-generated*); (2) conditioned on the prediction, it either produces structured anomaly annotations (for AI images) or provides a neutral content description (for real images). Across diverse real images, AnomReasonor consistently responds with "real" and generates detailed scene descriptions without hallucinating anomalies. This behavior demonstrates that the model does not indiscriminately project anomaly patterns learned from AIGC images onto real photos, and that its reasoning module remains stable even when anomaly supervision is absent. Qualitative

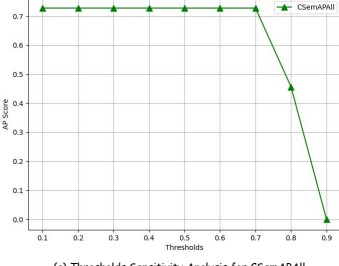 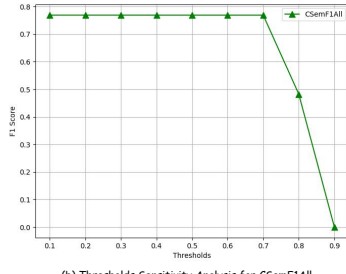

(a) Thresholds Sensitivity Analysis for CSemAPAll        (b) Thresholds Sensitivity Analysis for CSemF1All

Figure 10: **Threshold sensitivity analysis.** $CSemAP_{Full}$ (left) and $CSemF1_{Full}$ (right) under $\tau \in \{0.1, \ldots, 0.9\}$. Metrics are stable up to $\tau = 0.7$ and degrade only under excessively strict similarity thresholds ($\tau \geq 0.8$). The selected thresholds $\{0.7, 0.8, 0.9\}$ thus span stable, moderately strict, and failure regimes.

results further confirm that structured anomaly predictions emerge only when the model is confident the image is AI-generated, aligning detection with downstream explanation and highlighting the robustness of the two-step design.

### B.3 THRESHOLD SENSITIVITY ANALYSIS.

We evaluate the robustness of our semantic matching thresholds by sweeping $\tau \in \{0.1, \ldots, 0.9\}$ and reporting $CSemAP_{Full}$ and $CSemF1_{Full}$ (Fig. 10). Both metrics remain stable for $0.1 \leq \tau \leq 0.7$, indicating low sensitivity to moderate changes in the similarity cutoff. Performance drops only when $\tau \geq 0.8$, where matching becomes overly strict and penalizes natural linguistic variation. Our chosen thresholds $\{0.7, 0.8, 0.9\}$ therefore capture three regimes— stable (0.7), stricter (0.8), and near-failure (0.9)—providing a balanced measure of semantic alignment while maintaining robustness to minor paraphrasing differences.

### B.4 SCALING TO A STRONGER BASE MODEL

To validate that our training protocol and the AnomReason benchmark generalize beyond a single architecture size, we fine-tune a significantly stronger base vision-language model, **Qwen3-VL-8B**, using the same LoRA setup and training procedure.

Table 4 reports results on AnomReason-Deepfake. We compare against: (i) GPT-4o, the strongest closed-source system in our study; and (ii) AnomReasonor-7B (qwen2.5-vl-7B), our best-performing 7B model from the main paper.

Table 4: Performance of a stronger fine-tuned model (Qwen3-VL-8B) on AnomReason-Deepfake. The model surpasses both GPT-4o and our previous 7B AnomReasonor across all semantic metrics.

| Model | Acc | $CSemAP_{Phe}$ | $CSemAP_{Rea}$ | $CSemAP_{Full}$ | $CSemF1_{Phe}$ | $CSemF1_{Rea}$ | $CSemF1_{Full}$ |
|---|---|---|---|---|---|---|---|
| GPT-4o | 87.76 | 0.3750 | 0.3487 | 0.3612 | 0.4054 | 0.3770 | 0.3905 |
| AnomReasonor-7B (Setting 2) | 82.61 | 0.3684 | 0.3574 | 0.3613 | 0.4048 | 0.3929 | 0.3972 |
| AnomReasonor (Qwen3-8B) | **90.49** | **0.3890** | **0.3802** | **0.3832** | **0.4129** | **0.4034** | **0.4066** |

Scaling the base architecture from 7B to 8B yields consistent gains across all semantic evaluation metrics. Notably, *AnomReasonor(Qwen3-8B)* outperforms GPT-4o on both CSemAP and CSemF1, demonstrating that the benefits of our training framework are not tied to a specific model family or size. These results further confirm that AnomReason enables strong semantic reasoning capabilities when applied to modern VLMs.

## C   EVALUATION PROTOCOL DETAILS

This appendix provides a comprehensive description of our semantic evaluation protocol, used to score structured visual anomaly predictions as described in Sec. 3.3.

### C.1 SEMANTIC SIMILARITY: BERTSCORE CONFIGURATION

We instantiate all field-wise similarities with *BERTScore* (F1 variant). Unless noted, we freeze the backbone and preprocessing across all experiments to ensure comparability. Given hypothesis $h$ and reference $r$, we use the BERTScore F1:

$$\text{BERTScore}(h,r) = \frac{2\,P(h,r)\,R(h,r)}{P(h,r) + R(h,r)} \in [0,1], \tag{7}$$

and define

$$\begin{aligned}
\text{sim}_{\text{Phe}} &= \text{BERTScore}(\hat{o}, o), \\
\text{sim}_{\text{Rea}} &= \text{BERTScore}(\hat{r}, r), \\
\text{sim}_{\text{Full}} &= \alpha\,\text{sim}_{\text{Phe}} + (1-\alpha)\text{sim}_{\text{Rea}}.
\end{aligned}$$

The balanced mean for **Full** provides a smooth joint signal while discouraging over-optimizing a single field. We use $\alpha = 0.5$ unless otherwise stated.

### C.2 MATCHING AND CONTINGENCY COUNTS

For a view $v \in \{\text{Phe}, \text{Rea}, \text{Full}\}$ and threshold $\tau \in \{0.7, 0.8, 0.9\}$, define the indicator

$$T_v(\hat{a}, a; \tau) = \mathbb{1}\{\text{sim}_v(\hat{a}, a) \geq \tau\}. \tag{8}$$

**Ranking.** Sort predictions $\widehat{\mathcal{A}}(I) = \{\hat{a}_{(k)}\}_{k=1}^{K_I}$ by confidence $\hat{s}$ in descending order. **Assignment.** Scan $k = 1 \ldots K_I$; for each $\hat{a}_{(k)}$ choose the unmatched ground-truth $a^\star \in \mathcal{A}(I)$ maximizing $\text{sim}_v(\hat{a}_{(k)}, a)$ subject to $T_v = 1$. If such $a^\star$ exists, mark a true positive (TP) and lock both; else, mark a false positive (FP). Unmatched ground truths are false negatives (FN). **Tie-breaking.** If multiple $a$ achieve the same similarity, prefer the one with the highest $\text{sim}_{\text{Full}}$ (then smallest index). This one-to-one greedy rule prevents multi-matching and implicitly penalizes duplicates.

### C.3 PER-IMAGE AP/F1, PER-THRESHOLD AGGREGATION, DATASET AGGREGATION

For image $I$ at threshold $\tau$, let cumulative counts at rank $k$ be $\text{TP}_I(k, \tau)$ and $\text{FP}_I(k, \tau)$. Define

$$P_I(k, \tau) = \frac{\text{TP}_I(k, \tau)}{\text{TP}_I(k, \tau) + \text{FP}_I(k, \tau)}, \qquad R_I(k, \tau) = \frac{\text{TP}_I(k, \tau)}{|\mathcal{A}(I)|}. \tag{9}$$

**AP at $\tau$.**

$$\text{AP}_v(I, \tau) = \sum_{k:\,\text{new TP at } k} P_I(k, \tau)\,\big(R_I(k, \tau) - R_I(k{-}1, \tau)\big). \tag{10}$$

**Per-image SemAP (average over thresholds).**

$$\text{SemAP}_v(I) = \frac{1}{|\mathcal{T}|} \sum_{\tau \in \mathcal{T}} \text{AP}_v(I, \tau), \quad \mathcal{T} = \{0.7, 0.8, 0.9\},\ |\mathcal{T}| = 3. \tag{11}$$

**Dataset SemAP (macro over images).**

$$\text{SemAP}_v = \frac{1}{|\mathcal{D}|} \sum_{I \in \mathcal{D}} \text{SemAP}_v(I). \tag{12}$$

**Per-image F1 at $\tau$.**

$$\text{F1}_v(I, \tau) = \frac{2\,P_I(\tau)\,R_I(\tau)}{P_I(\tau) + R_I(\tau)}, \quad P_I(\tau) = \frac{\text{TP}_I}{\text{TP}_I + \text{FP}_I}, \quad R_I(\tau) = \frac{\text{TP}_I}{|\mathcal{A}(I)|}. \tag{13}$$

**Dataset SemF1.** We report the simple average across images and thresholds:

$$\text{SemF1}_v = \frac{1}{|\mathcal{D}|\,|\mathcal{T}|} \sum_{I \in \mathcal{D}} \sum_{\tau \in \mathcal{T}} \text{F1}_v(I, \tau), \tag{14}$$

with per-threshold breakdowns provided below.

In Sec. 4.2, we introduce *classification-aware* variants of our semantic metrics, denoted as $\text{CSemAP}_v$ and $\text{CSemF1}_v$ for $v \in \{\text{Phe}, \text{Rea}, \text{Full}\}$. Let $y(I) \in \{\text{real}, \text{AI}\}$ be the ground-truth source and $\hat{y}(I)$ the predicted source. For a view $v \in \{\text{Phe}, \text{Rea}, \text{Full}\}$ and threshold set $\mathcal{T} = \{0.7, 0.8, 0.9\}$, define

$$\text{CSemAP}_v = \frac{1}{|\mathcal{D}||\mathcal{T}|} \sum_{I \in \mathcal{D}} \sum_{\tau \in \mathcal{T}} \mathbb{1}\!\!\!\!/\{\hat{y}(I) = y(I)\} \, \text{AP}_v(I, \tau), \tag{15}$$

$$\text{CSemF1}_v = \frac{1}{|\mathcal{D}||\mathcal{T}|} \sum_{I \in \mathcal{D}} \sum_{\tau \in \mathcal{T}} \mathbb{1}\!\!\!\!/\{\hat{y}(I) = y(I)\} \, \text{F1}_v(I, \tau), \tag{16}$$

Explanations are scored only when detection is correct, and assigned zero otherwise. This ties explanation quality to valid classification, discouraging post-hoc rationalization for mispredictions and promoting joint interpretability.

# D  TRAINING DETAILS

## D.1  DETAILS FOR SEC.4.1

We fine-tune Qwen2.5-VL-7B using LoRA modules inserted into the multimodal attention blocks, while keeping the visual encoder frozen. Specifically, we adopt the official Qwen2.5-VL-7B backbone (7B parameters) with its native image projector, freezing the vision tower (ViT). LoRA is applied to the attention projections (q_proj, k_proj, v_proj) at every 4th transformer layer, with a rank of 8, scaling factor $\alpha = 16$, and dropout rate of 0.5. Fine-tuning is conducted for one epoch using a batch size of 4 and gradient accumulation of 8 on 4 A6000 GPUs. Input resolution is dynamically adjusted with a minimum of 50,176 pixels and a maximum of 614,656 pixels.

**Supervision data and construction**   We use the official train split of the *AnomReason* dataset for training, strictly retaining only *verified* anomalies to ensure label quality. For each image $I$, we construct one or multiple training instances depending on sequence length constraints. When an image contains $m$ verified anomalies $\{(y_j, o_j, r_j, v_j)\}_{j=1}^{m}$, we pack them into a single instance in randomized order unless token limits require splitting.

```
Human:  Analyze the provided AI-generated image to
detect all semantic anomalies.  Provide a detailed
list of anomalies using the following structure:
- @1.  Name:  [Phenomenon Name] - Phenomenon:
[Description of the visible anomaly] - Reasoning:
[Step-by-step explanation of why this phenomenon is
unrealistic or illogical] - Severity Score:  [Rate
the issue from 0 (completely unnatural) to 100 (fully
realistic)]
```

The model is expected to produce responses such as:

```
@1.  Name:  Woman's shirt fabric rigidity
Phenomenon:  The shirt worn by the woman appears
overly smooth and rigid, showing no folds, creases,
or deformation where it interacts with the baby or the
reclining posture.
Reasoning:  Soft fabrics typically conform to the body
and surrounding objects, creating visible folds and
compression due to gravity and physical contact.  The
lack of such effects reduces the plausibility of the
material behavior.
Severity Score:  20.

@2.  Name:  Pillow compression inconsistency
Phenomenon:  The pillow supporting the woman's head
```

```
and shoulder appears unnaturally smooth and rigid,
showing no visible deformation or compression despite
the weight applied.
Reasoning:  Pillows are designed to compress under
pressure, creating indentations and folds where
weight is applied.  The lack of deformation suggests
unrealistic material behavior, detracting from the
plausibility of the scene.
Severity Score:  25.
```

This format ensures consistent alignment between training supervision and evaluation, facilitating learning of structured anomaly descriptions and reasoning patterns. The use of step-wise justifications and severity scoring further supports interpretable and rank-aware detection.

### D.2   DETAILS FOR SEC.4.2

We fine-tune **Qwen2.5-VL-7B** using low-rank adapters (LoRA), following the same hyperparameters described in Appendix D.1, with the exception that we set the dropout rate for LoRA (`lora_dropout`) to 0.9 for this setting.

**Turn 1: Source classification.**   The first turn asks the model to determine the image source in binary form:

```
Q: Summarize whether this image is Generated by
Artificial Intelligence, please only return yes or
no.
A: <Yes, this image is generated by AI | No, this
image is a real photograph.>
```

**Turn 2: Semantic anomaly explanation.**   If the image is AI-generated, the second turn prompts the model to enumerate all semantic-level inconsistencies using a structured output format:

```
Q: If AI-generated, list semantic anomalies:
Name, Phenomenon, Reasoning, Severity.
A: [
{Name:  <y1>, Observed:  <o1>, Reasoning:  <r1>,
Severity:  <v1>},
...
]
```

If the image is labeled as real, the second answer defaults to a content-based explanation of what is observed in the image without marking any anomaly. This dual-turn QA formulation enables the model to explicitly separate generation source classification from semantic inconsistency identification, promoting robust deepfake understanding and explainability.

## E   ADDITIONAL DETAILS ON ANOMALY DETECTION AGENTS

In this section, we provide detailed insights into the design and implementation of the anomaly detection agents used in our framework. Each agent plays a crucial role in analyzing different aspects of the input image, allowing for an in-depth semantic analysis of AI-generated images. The design of these agents is driven by the need to model high-level perceptual reasoning and commonsense logic, which are essential for detecting complex semantic anomalies that would otherwise go unnoticed by traditional anomaly detection methods.

### E.1   OBJECT PERCEIVER AGENT

The **ObjectPerceiver** agent is tasked with identifying and parsing all semantically distinct entities within an input image. This is the first step in the anomaly detection pipeline, as it helps isolate the

relevant objects in the image for further analysis. By detecting objects in the image, the agent serves as the foundation for all subsequent analysis.

**Design Motivation:** The motivation behind the Object Perceiver is to provide a structured identification of objects in the image, which can then be used for deeper anomaly detection in subsequent steps. The design intentionally focuses on high-level object semantics rather than low-level features like lighting or texture inconsistencies, which might not be as impactful for detecting semantic anomalies.

**Task:** Analyze all objects and individuals in the image. For each object or individual, provide a detailed, accurate, and comprehensive description, while identifying any inconsistencies, anomalies, or illogical aspects. Ensure no object or body part is omitted.

**Follow the steps below and provide your analysis in the structured format specified:**

- Identify and describe all objects and individuals in the image.
- For each object or individual, provide a detailed, accurate, and comprehensive description.
- Highlight any inconsistencies, anomalies, or illogical aspects in:
  - **Shape and Structure:** Are there distortions, missing parts, or unnatural forms?
  - **Material and Texture:** Are there abrupt texture changes or mismatches?
  - **Lighting and Shadows:** Are the lighting and shadows consistent with the environment?
  - **Physical Properties:** Are there any violations of real-world physics or logic (e.g., floating objects)?
  - **Common Sense Verification:** Are there any semantic inconsistencies (e.g., a door handle on a chair)?
  - **Human Anatomy (if applicable):** Identify unnatural features such as missing limbs, extra fingers, or disproportionate body parts.

**Output Format:**
Each object/body part should be described individually in the following structured format:

#Name#: Detailed Description.

#Name#: Detailed Description.

**Example Output:**

Person: A man with three arms, one of which is unnaturally attached to his back. He wears a blue jacket.

Chair: A wooden chair that appears to be floating without support, casting no shadow.

Dog: A golden retriever with two tails, one of which is blurry and semi-transparent.

```
Highlight all implausible, unnatural, or inconsistent
details while ensuring full coverage of the image
content.  Only output the list in the specified
format.
```

**Effectiveness:** This prompt ensures that the Object Perceiver focuses on capturing the main objects and entities in the image while disregarding irrelevant details. By emphasizing the semantics of each object, the agent avoids common pitfalls of pixel-based anomaly detection and lays the groundwork for higher-level analysis.

### E.2 ATTRIBUTE ANALYZER AGENT

The **Attribute Analyzer** is responsible for detecting anomalies in the visual attributes of objects, such as shape, texture, material, and other intrinsic properties. This agent examines internal inconsistencies within the objects themselves and identifies any attributes that deviate from the expected real-world norms.

**Design Motivation:** The rationale behind the Attribute Analyzer is that many semantic anomalies manifest as inconsistencies in object attributes. For example, an object might appear out of place because of an unusual shape or texture that violates expectations based on the scene context. By focusing on attributes, this agent can detect low-level anomalies that might not be caught by higher-level reasoning alone.

**AttributeAnalyzer Step1 Prompt:**

```
Task:  Analyze {Current object} in the image.

Focus on analyzing and identifying any anomalies in
the following aspects:
  1. Shape and Structure
     • Are there unnatural forms or distortions?
     • Are proportions consistent with the object's
       design?
  2. Functionality
     • Does the object behave logically in real-world
       scenarios?
     • Are there physical impossibilities (e.g.,
       unsupported structures)?
  3. Human Body Structure Verification (if applicable)
     • Are limbs, fingers, and facial features
       correctly placed and proportional?
     • Are there unnatural fusions, duplications, or
       disconnections?

  Deliverable:
     • Highlight all implausible, unnatural, or
       inconsistent details.
     • Ensure a thorough analysis that covers all aspects
       of the image content.
     • Provide concise, evidence-based explanations for
       all findings.
```

**AttributeAnalyzer Step2 Prompt:**

```
Object:  {Current object}
Description Input:  {AttributeAnalyzer Step1 Response}

Task:  Analyze the detailed description of {Current
object} and identify all unreasonable, contradictory,
```

```
or physically impossible details specific to this
object.
```

**Provide a structured list of issues using the following format:**
- **Abnormal Phenomenon Name:** The name of the observed anomaly.
- **Observed Issue:** The unnatural feature found.
- **Explanation:** Why this characteristic is unrealistic.

**Example Output:**
1. **Abnormal Phenomenon Name:** Streetlight No Power
   **Observed Issue:** The streetlight is glowing but has no power source or wiring.
   **Explanation:** A streetlight requires an electrical connection to function, and no wires or batteries are visible.

**Instructions:**
- Analyze **only** {Current object}.
- Output **only** issues directly related to {Current object}, using the specified format.

**Effectiveness:** This two-stage approach encourages detailed yet structured interpretation of each object's attributes. The separation of observation and reasoning mimics human perceptual processes and aligns well with structured evaluation protocols introduced in Sec. 3.3.

### E.3 RELATION REASONER AGENT

The **RelationReasoner** analyzes spatial, semantic, and functional relationships between objects in the image. It checks if objects are interacting in a plausible way or if their relationships defy physical or logical laws. This agent is critical for detecting anomalies that arise from the interaction between objects.

**Design Motivation:** The design of the Relation Reasoner is motivated by the fact that many semantic anomalies stem from the way objects relate to one another. For example, objects that should interact, such as a hand holding a cup, may appear to be floating or not touching each other at all. By modeling relationships, the agent detects high-level semantic inconsistencies that are often missed by attribute analysis alone.

**RelationReasoner Step1 Prompt:**

**Task:** Analyze the spatial and logical relationships between **{Current object}** and the following objects: ({*all objects from ObjectPerceiver*}).
You should evaluate:
- One-to-one relationships (e.g., **{Current object}** with each object)
- One-to-many relationships (e.g., **{Current object}** in relation to multiple objects collectively)

**Context Descriptions:**
{Object Descriptions from AttributeAnalyzer}

**Focus Areas:**
1. **Perspective Errors:** Are objects placed in impossible or illogical locations relative to **{Current object}**?

2. **Physical Interactions:** Are there unnatural interactions (e.g., floating without support, overlapping unnaturally)?
3. **Logical Contradictions:** Are there contradictions with real-world behavior or common-sense logic?

**Instructions:**

- Focus analysis on **{Current object}** as the primary subject.
- For **one-to-one relationships,** evaluate individual pairings.
- For **one-to-many relationships,** consider collective spatial, logical, and contextual coherence.

**Output Format** (structured report for each issue):

- **Relationship:** Describe the relationship being analyzed.
- **Observed Issue:** Detail the anomaly or inconsistency.
- **Explanation:** Explain why the issue is illogical or unrealistic.

**Deliverables:**

- Analyze all one-to-one and one-to-many relationships involving **{Current object}**.
- Ensure detailed reasoning and structured output for each detected issue.

**RelationReasoner Step2 Prompt:**

**Relation Input:** {RelationReasoner Step1 Response}

**Focus Object:** The primary subject of analysis is **{Current object}**. All evaluations should center on **{Current object}** and its relationships with the following objects: ({all objects from ObjectPerceiver}).

**Task:** Based on the prior relationship analysis, analyze and summarize the relationships between **{Current object}** and the listed objects. Emphasize detection of logical contradictions, physical impossibilities, and semantic anomalies.

**Key Aspects to Evaluate:**

1. **Logical Coherence:** Are the relationships internally consistent?
   - Example: An object cannot be both inside and outside another simultaneously.
2. **Physical Realism:** Do the relationships conform to real-world physical laws?
   - Example: Objects should not float without visible support.
3. **Semantic Plausibility:** Are the interactions meaningful and contextually appropriate?

```
        • Example:  A dog \wearing" a cloud is not
          semantically plausible.
     4. Causal Consistency:  Do object states logically
        follow from their relationships?
        • Example:  A book balanced on a steep slope
          should be expected to fall.

    Output Format:  For each detected anomaly, provide a
    structured report as follows:
        • Objects Involved:  List the relevant objects
          (including {Current object}).
        • Observed Issue:  Describe the logical, physical,
          or semantic anomaly.
        • Reasoning:  Justify why this relationship is
          unnatural, implausible, or illogical.

    Instructions:
        • Focus exclusively on {Current object} and its
          relationships.
        • Evaluate both individual (one-to-one) and group
          (one-to-many) relationships.
```

**Effectiveness:** By evaluating the spatial and functional relations between objects, the Relation Reasoner helps detect anomalies that would violate everyday common sense or physical laws. For example, it would catch a cup floating in mid-air without any support or a person walking through a solid wall. This agent significantly enhances the ability to detect high-level semantic anomalies that go beyond object-level properties.

E.4 ANOMALY INTEGRATION AND FORMATTING

After the anomalies are detected by the Attribute Analyzer and Relation Reasoner, they are passed to the **Anomaly Integration** and **Anomaly Formatting** agents. The Integration agent consolidates similar or redundant anomalies and eliminates noise, while the Formatting agent structures the anomalies into a final output format.

**Design Motivation:** The design of the Anomaly Integration and Formatting agents is driven by the need to ensure that the detected anomalies are presented in a clear and interpretable manner. The Integration agent helps combine similar anomalies into one, while the Formatting agent ensures the final output is structured and easy to use for downstream applications like quality assessment or dataset debugging.

**AnomalyIntegrator Step1 Prompt:**

```
    Description for {Current object}:  {AttributeAnalyzer
    Response}
    Relationships of {Current object} with other
    objects ({all objects from ObjectPerceiver}):
    {RelationReasoner Response}

    Task:  Review and analyze the detailed Description
    and Relationships of {Current object}.  Summarize
    all unreasonable, contradictory, or physically
    impossible details related to {Current object}, while
    consolidating similar or repeated anomalies into a
    comprehensive report.

    Focus Areas:
```

1. **Contradictory Details:** Identify conflicting statements or relationships (e.g., "floating" vs. "resting on the ground").
2. **Unnatural Behaviors:** Highlight features or actions implausible in real-world settings.
3. **Spatial Inconsistencies:** Detect impossible locations or orientations for **{Current object}** or others.
4. **Illogical Physical Properties:** Point out violations of physics or reality (e.g., water flowing upward).

**Instructions:**

- Consolidate similar anomalies from both **Description** and **Relationships**.
- Center all findings around **{Current object}** and its interactions with other objects.

**Output Format** (structured list):

1. **Observed Phenomenon:** *Brief summary of the inconsistency*
   - **Sources:** Indicate if the issue comes from the **Description, Relationships,** or **Both.**
   - **Details:** Provide specific observations related to the anomaly.
   - **Explanation:** Justify why the phenomenon is contradictory, unnatural, or implausible.

**Deliverables:**

- Focus exclusively on **{Current object}**.
- Consolidate and summarize issues across **Description** and **Relationships**.
- Output only the structured list in the specified format.

**AnomalyIntegrator Step2 Prompt:**

**Anomalies:** {AnomalyIntegrator Step1 Response}

**Task:** Summarize and categorize all detected unnatural, illogical, or inconsistent phenomena in the image.

**For each issue, provide:**

1. **Object Name:** Clearly identify the object(s) involved.
2. **Phenomenon:** Describe the unnatural or illogical aspect of the object(s).
3. **Explanation:** Explain why this phenomenon is unrealistic, referencing real-world physics, anatomy, perspective, or common sense.

**Output Format:** Provide a structured list using the format below:

**Example Output:**

1. **Object Name:** Tree
   **Phenomenon:** The tree trunk bends at an impossible 90-degree angle.
   **Explanation:** Real trees cannot grow in this shape due to gravitational constraints.
2. **Object Name:** Dog
   **Phenomenon:** The dog has three tails, one of which is semi-transparent.
   **Explanation:** This is anatomically impossible for dogs.

**Instructions:**
- Only output the list in the specified format.
- Ensure each anomaly is clearly tied to a specific object.
- Exclude unrelated content or commentary.

### AnomalyFormatter Prompt:

**The following are a list of pre-selected anomalies:**
{List of AnomalyIntegrator Step2 Response}

**Task:** From the list above, identify and summarize the **visually prominent and semantically significant anomalies** observed in the image.
You must analyze, consolidate, and explain each anomaly in a way that is **logical, detailed, and persuasive**, as if communicating to both experts and non-experts.

**Instructions:**

1. **Merge Similar or Redundant Anomalies**
   - Group phenomena sharing a common cause, concept, or visual effect.
   - Avoid repetition by merging entries describing the same core issue.
2. **Resolve Contradictions Thoughtfully**
   - If descriptions conflict, reconcile them using physical laws, biological plausibility, and visual logic.
   - Summarize both viewpoints if both are partially valid.
3. **Filter Out Non-Visible or Insignificant Issues**
   - Omit anomalies that are not visually apparent (e.g., minor texture noise).
   - Focus on what is **clearly and prominently visible**.
4. **Justify with Real-World Logic**
   - Support each anomaly with logical, physical, anatomical, or functional reasoning.
5. **Do Not Parrot the Input**
   - Rephrase and reinterpret anomalies based on visual evidence and contextual understanding.
6. **Ensure Coverage**
   - All input anomalies must be included, either directly or through consolidation.

```
Output Format:  Write a numbered list.  For each
entry, use the following structure:

@1. **Name**: [Descriptive title of the anomaly]
- **Observed Phenomenon**:
    - Describe what is visibly wrong in visual terms.
    - Include positions, shapes, textures, or contextual oddities.
    - Ensure clarity without needing to see the image.
- **Reasoning**:
    - Explain why this is implausible.
    - Support with physical laws, anatomy, real-world logic,
    or social context.
- **Severity Score**: [0{100; 0 = fully unrealistic,
    100 = fully realistic]
```

**Example Output:**

```
 1. Name:  Abnormal number of hands
    Observed Phenomenon:  The individual on the left
    has two left hands, one emerging from the elbow
    and overlapping with the sleeve.  Both hands
    share identical orientation and lack anatomical
    continuity.
    Reasoning:  Human anatomy allows one left
    and one right hand.  Two left hands in such
    arrangement violate biological symmetry and visual
    plausibility.
    Severity Score:  5/100 (highly unrealistic)
 2. Name:  Suspended chair without support
    Observed Phenomenon:  A wooden chair is floating
    approximately 30 cm above the ground without
    visible support or shadows.
    Reasoning:  Gravity requires contact or
    suspension; absence of legs, shadows, or wires
    defies physical realism.
    Severity Score:  10/100 (extremely unnatural)
```

**Final Notes:**

- Output only the structured list in the format above.
- Think critically.  Be precise, complete, and persuasive.
- Provide a human-understandable summary of core visual anomalies in the image.

**Effectiveness:** These final stages ensure that the output of the anomaly detection process is both precise and interpretable. The integration step reduces redundancy and focuses on the most significant anomalies, while the formatting step makes it easier to present and use the results for further analysis.

**Overall Design and Significance:** The design of the anomaly detection agents is driven by the need for both local and global anomaly detection in AI-generated images. By breaking down the process into three main agents—Object Perceiver, Attribute Analyzer, and Relation Reasoner—each agent can focus on a specific aspect of the image, from identifying objects and their attributes to analyzing how they interact in a scene. This modular approach allows for greater flexibility and interpretability in the detection process. Furthermore, the integration and formatting stages ensure that the final output is structured and usable for real-world applications.

The multi-agent design enables a more nuanced and robust detection of semantic anomalies, making the framework capable of handling complex image generation models like those used in deepfake detection and other high-level image assessments.

## F    COMPARATIVE ANALYSIS: HUMAN VS. ANOMAGENT ANNOTATIONS

To evaluate the effectiveness and consistency of our proposed anomaly detection pipeline, we conducted a comparative study between **AnomAgent** and **human annotators**. We used a benchmark set of **1,000 AI-generated images** with rich semantic content and potential visual inconsistencies.

### F.1    ANNOTATION TASK AND SETUP

Each image was annotated independently by:

- **Human annotators**, we select several annotators with advanced academic backgrounds in computer vision, specifically individuals holding master's or doctoral degrees..
- **AnomAgent**, using the pipeline described in Appendix E.

Annotators were instructed to identify any *implausible, inconsistent, or unnatural visual features*, along with a brief description of the issue. AnomAgent, in contrast, automatically generated structured reports with descriptions, reasoning, and severity.

### F.2    QUALITATIVE COMPARISON

Figure 11 shows side-by-side comparisons of annotations from AnomAgent and human annotators. Each example highlights different strengths:

- **Fine-grained detection**: AnomAgent captured subtle anomalies (e.g., missing shadows, unnatural finger joints) often missed by humans.
- **Consistency**: Across images with similar anomalies (e.g., floating objects), AnomAgent produced structurally similar descriptions, whereas human annotations varied in terminology and detail.
- **Reasoning support**: Human annotators often provided short descriptions without justification, while AnomAgent included structured, interpretable explanations.

For a dataset comprising 1000 images, human annotators identified 4884 anomalies, whereas AnomAgent detected 8290 anomalies. AnomAgent not only detects more anomalies, but does so with higher consistency and full interpretability with structured anomaly description.

## G    COMPARISON OF BERTSCORE BACKBONES FOR SEMANTIC EVALUATION

Our semantic evaluation framework (Section 3.3) relies on BERTScore to compare structured textual anomaly explanations. However, BERTScore's output depends heavily on the underlying language model used to compute embeddings. To assess the stability of our evaluation metrics, we benchmark several pre-trained language models as BERTScore backbones. This analysis tests whether the relative performance trends of vision-language models (VLMs) hold across different similarity functions.

In addition to the default `distilbert-base-uncased` model used in the main paper, we evaluate two widely used and diverse backbones:

- `roberta-large-mnli`
- `google/mt5-large`

We use the exact same evaluation pipeline as in Section 3.3, computing SemAPand SemF1metrics between human-verified ground truth anomalies and predictions from each model.

Table 5 and Table 6 present semantic detection and reasoning performance for all evaluated models across both backbones.

The absolute values of SemAPand SemF1vary across backbones. For example, `mt5-large` consistently yields lower scores due to its multilingual and less fine-grained English representations.

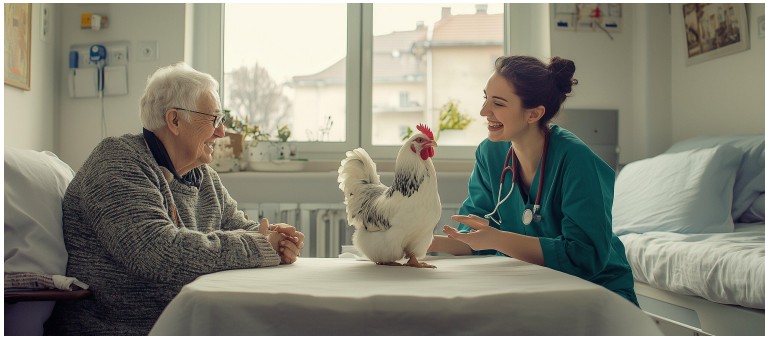

1. **Chicken feet are unrealistic**: The anatomy of the chicken's feet appears to be incorrect or not convincing, which could suggest that it is an artificial creation lacking an accurate depiction of avian anatomy.

2. **Hand abnormalities:** The hands or fingers in the image might have distortions, irregular shapes, or inconsistent properties that do not align with natural human anatomy, indicating AI-generation imperfections.

(a) Human Annotation

**Name: Presence of a chicken in a medical consultation setting**

**Phenomenon:** A chicken is positioned on a table in a clinical environment, surrounded by medical furniture and individuals who appear to be engaged in a discussion. The chicken is treated as though it is part of the consultation.

**Reasoning:** Chickens are farm animals and are not typically present in sterile medical environments. Their inclusion in this setting contradicts hygiene standards and the expected purpose of the space, which is designed for human healthcare. The lack of veterinary-specific tools further emphasizes the implausibility of the scene.

**Severity Score:** 20

**Name: Lack of reaction from individuals to the chicken**

**Phenomenon:** The elderly person and the medical professional do not display any visible reaction to the presence of the chicken, despite its unusual placement and context. Their body language suggests they are engaged in a normal conversation, ignoring the chicken entirely.

**Reasoning:** In real-world scenarios, the presence of a chicken in a clinical environment would likely provoke curiosity, surprise, or interaction from the individuals present. The lack of acknowledgment creates a semantic disconnect and undermines the plausibility of the scene.

**Severity Score:** 25

**Name: Use of a tablecloth in a clinical setting**

**Phenomenon:** The table is covered with a cloth, which is unusual for a medical environment. Clinical tables are typically designed with easy-to-clean surfaces to maintain hygiene.

**Reasoning:** The use of a tablecloth contradicts standard practices in medical settings, especially in the presence of an animal, which could introduce contaminants. This detail undermines the functional coherence of the scene.

**Severity Score:** 15

(b) AnomAgent Annotation

Figure 11: Comparison of human annotations and AnomAgent outputs. AnomAgent identifies more anomalies with detailed explanations.

In contrast, `roberta-large-mnli` produces more moderate values, balancing semantic sensitivity with surface-level alignment. However, the relative ranking of models—e.g., GPT-4o, AnomReasonor-7B, and Qwen2.5 variants—remains largely unchanged, confirming the robustness of our evaluation findings.

Metrics related to reasoning quality (SemAP$_{Rea}$, SemF1$_{Rea}$) show greater variation across backbones than observation-only metrics. This aligns with the expectation that free-form reasoning texts introduce more lexical and structural variability, making similarity estimation more dependent on model semantics.

Across both backbones, *AnomReasonor-7B* achieves top-tier performance, closely tracking or surpassing proprietary models on reasoning-related submetrics. This reinforces its strong generalization in structured explanation tasks and validates the effect of fine-tuning on our anomaly supervision.

While the choice of BERTScore model influences raw scores, the relative performance trends among models remain stable. Thus, our benchmark conclusions are not overly sensitive to the specific similarity function used—providing evidence of metric robustness and cross-backbone reliability.

Table 5: Performance comparison using `roberta-large-mnli` as BERTScore backbone.

| Model | SemAP$_{Phe}$ | SemAP$_{Rea}$ | SemAP$_{Full}$ | SemF1$_{Phe}$ | SemF1$_{Rea}$ | SemF1$_{Full}$ |
|---|---|---|---|---|---|---|
| LongVA-7B | 0.2243 | 0.2002 | 0.2176 | 0.1372 | 0.1185 | 0.1325 |
| LLaVA-OV-7B | 0.2487 | 0.2107 | 0.2358 | 0.1514 | 0.1269 | 0.1435 |
| Phi-3.5-Vision | 0.2427 | 0.2129 | 0.2324 | 0.2074 | 0.1798 | 0.1984 |
| MiniCPM-V-2.6 | 0.2949 | 0.2621 | 0.2844 | 0.1417 | 0.1244 | 0.1366 |
| InternVL2-8B | 0.2620 | 0.2297 | 0.2512 | 0.2674 | 0.2346 | 0.2563 |
| InternVL2.5-8B | 0.2455 | 0.2079 | 0.2320 | 0.2179 | 0.1843 | 0.2062 |
| InternVL3-8B | 0.3320 | 0.2811 | 0.3127 | 0.1405 | 0.1190 | 0.1332 |
| InternVL3-9B | 0.2697 | 0.2313 | 0.2507 | 0.2738 | 0.2352 | 0.2542 |
| mPLUG-Owl3-7B | 0.3102 | 0.2768 | 0.2984 | 0.0965 | 0.0857 | 0.0928 |
| Qwen2-VL-7B | 0.3156 | 0.2749 | 0.3006 | 0.1032 | 0.0914 | 0.0989 |
| InternVL2-26B | 0.2876 | 0.2532 | 0.2727 | 0.2760 | 0.2434 | 0.2621 |
| Qwen2.5-VL-7B | 0.3190 | 0.2791 | 0.3017 | 0.2893 | 0.2531 | 0.2739 |
| Qwen2.5-VL-72B | 0.3379 | 0.2906 | 0.3125 | 0.3033 | 0.2607 | 0.2809 |
| Gemini-2.5-pro | 0.2485 | 0.2290 | 0.2360 | 0.1330 | 0.1221 | 0.1264 |
| GPT-o3 | 0.2860 | 0.2578 | 0.2860 | 0.2834 | 0.2555 | 0.2834 |
| GPT-5 | 0.2322 | 0.2156 | 0.2207 | 0.2718 | 0.2526 | 0.2582 |
| GPT-4o | 0.3434 | 0.2955 | 0.3083 | 0.3707 | 0.3188 | 0.3325 |
| AnomReasonor-7B | 0.3342 | 0.3197 | 0.3151 | 0.3243 | 0.3103 | 0.3059 |

Table 6: Performance comparison using `google/mt5-large` as BERTScore backbone.

| Model | SemAP$_{Phe}$ | SemAP$_{Rea}$ | SemAP$_{Full}$ | SemF1$_{Phe}$ | SemF1$_{Rea}$ | SemF1$_{Full}$ |
|---|---|---|---|---|---|---|
| LongVA-7B | 0.0535 | 0.0635 | 0.0393 | 0.0315 | 0.0331 | 0.0214 |
| LLaVA-OV-7B | 0.0993 | 0.1044 | 0.0834 | 0.0613 | 0.0599 | 0.0500 |
| Phi-3.5-Vision | 0.1326 | 0.0853 | 0.0943 | 0.1092 | 0.0666 | 0.0743 |
| MiniCPM-V-2.6 | 0.1289 | 0.0926 | 0.0884 | 0.0598 | 0.0426 | 0.0407 |
| InternVL2-8B | 0.1224 | 0.1243 | 0.1076 | 0.1252 | 0.1243 | 0.1089 |
| InternVL2.5-8B | 0.1029 | 0.1024 | 0.0869 | 0.0911 | 0.0844 | 0.0743 |
| InternVL3-8B | 0.1642 | 0.1764 | 0.1488 | 0.0661 | 0.0706 | 0.0596 |
| InternVL3-9B | 0.1391 | 0.1121 | 0.1125 | 0.1428 | 0.1125 | 0.1143 |
| mPLUG-Owl3-7B | 0.1173 | 0.1267 | 0.0964 | 0.0350 | 0.0379 | 0.0287 |
| Qwen2-VL-7B | 0.1306 | 0.1393 | 0.1136 | 0.0403 | 0.0427 | 0.0348 |
| InternVL2-26B | 0.1774 | 0.1417 | 0.1488 | 0.1701 | 0.1341 | 0.1420 |
| Qwen2.5-VL-7B | 0.2034 | 0.1697 | 0.1775 | 0.1839 | 0.1531 | 0.1605 |
| Qwen2.5-VL-72B | 0.2078 | 0.1798 | 0.1850 | 0.1864 | 0.1611 | 0.1662 |
| Gemini-2.5-pro | 0.1905 | 0.1402 | 0.1632 | 0.0989 | 0.0718 | 0.0839 |
| GPT-o3 | 0.1811 | 0.0991 | 0.1330 | 0.1793 | 0.0973 | 0.1311 |
| GPT-5 | 0.1724 | 0.0999 | 0.1346 | 0.2031 | 0.1177 | 0.1587 |
| GPT-4o | 0.2708 | 0.2171 | 0.2387 | 0.2933 | 0.2352 | 0.2586 |
| AnomReasonor-7B | 0.2736 | 0.2612 | 0.2653 | 0.2655 | 0.2532 | 0.2574 |

