# OpenReview forum: "Semantic Visual Anomaly Detection and Reasoning in AI-Generated Images"
_ICLR.cc/2026/Conference — ICLR 2026 Poster_

### Official Review · Reviewer_EDEz · 2025-10-25

**Soundness:** 2
**Presentation:** 3
**Contribution:** 2
**Rating:** 6
**Confidence:** 3

**Summary:**

This paper proposes AnomReason, a large-scale benchmark annotated with structured quadruples to capture commonsense, physical, and relational inconsistencies in generated images. The dataset is built using AnomAgent, a modular multi-agent framework that decomposes anomaly detection into entity parsing, anomaly mining, and structured output generation, with lightweight human-in-the-loop verification. The author also provides AnomReasonor-7B, a model fine-tuned on AnomReason, which achieves comparable performance with GPT-4o.

**Strengths:**

1. The paper is clearly written and well-structured.
2. The introduction of AnomReason, AnomReason Deepfake, and AnomAgent provides both a comprehensive benchmark and a scalable annotation framework for semantic-level anomaly detection.

**Weaknesses:**

1. The evaluations in Sections 4.1 and 4.2 are limited to the proposed AnomReason and AnomReason Deepfake benchmark. The paper should include results on additional third-party anomaly detection datasets, such as FakeReasoning, Spot the Fake, etc.
2. The evaluations in Sections 4.1 and 4.2 primarily evaluate general-purpose VLMs rather than models specifically designed or fine-tuned for anomaly or forgery detection, such as FakeReasoning, Spot the Fake, Aigi-holmes, etc.
3. The proposed pipeline mainly relies on prompting and LoRA fine-tuning. There is limited method innovation in model architecture or training strategy, which somewhat reduces the technical novelty of the work.

**Questions:**

1. Could you further analyze the results to determine what kinds of anomalies current T2I models are most likely to generate?
2. Given that AnomReasonor-7B shows the strongest anomaly detection ability, could you analyze why its performance is inferior to GPT-4o on AnomReason-Deepfake? Is it because not all AI-generated images in this benchmark exhibit visible anomalies?
3. How will the selection of hyperparameters in SFT influence AnomReasonor-7B’s performance?

---

> ### Author Response · Authors · 2025-11-24
> **Response to Reviewer EDEz**
>
> We thank you for the careful review. We are glad that you found the paper clearly written and well structured, and that you see AnomReason, AnomReason-Deepfake, and AnomAgent as providing a valuable benchmark and scalable annotation framework for semantic-level anomaly detection.
>
>
> **[W1-3] Scope of evaluation and architectural novelty; AnomReasonor as a baseline.**
>
> **A1:** Thank you for these thoughtful comments and for pointing out the potential value of evaluating on additional third-party datasets and specialized forgery detectors. We fully agree that cross-dataset evaluation and comparisons to dedicated models would further strengthen the empirical story.
>
> In this submission, our primary goal is to introduce and validate the AnomReason benchmarks and the associated task formulation, rather than to propose a highly engineered model. Accordingly, AnomReasonor is intentionally designed as a **simple, reproducible baseline**: we apply lightweight LoRA adapters to Qwen2.5-VL-7B (and Qwen3-VL-8B in our new experiments), without additional architectural tricks.
>
> Our current focus is on:
> - Defining the **semantic anomaly detection and reasoning** task;
> - Designing the **AnomAgent** modular multi-agent pipeline for structured annotation;
> - Building the **AnomReason** benchmarks with quadruple annotations and HITL verification; and
> - Providing semantics-aware metrics (**SemAP / SemF1**) tailored to anomaly-level evaluation.
>
> Many existing anomaly/forgery datasets and specialized models are not directly configured to output the structured quadruples required by our metrics, so integrating them in a fully comparable way would require substantial additional engineering and space that we unfortunately do not have for this submission. We therefore see AnomReasonor as a starting point rather than a final model, and we will clarify this positioning in the paper.
>
> That said, we very much appreciate your suggestion: one of our planned next steps is to adapt our semantic metrics and evaluation protocol to third-party datasets such as FakeReasoning and Spot the Fake, and to benchmark specialized detectors within this framework. We will make this future direction explicit in the revision.
>
> **[Q1] Further analysis: what kinds of anomalies current T2I models tend to generate.**
>
> **A2:** Thank you for this excellent suggestion. In the revised paper and appendix, we provide a more detailed analysis of the anomalies mined from AnomReason, from the perspective of text-to-image (T2I) model behavior. Our findings show that:
>  - Interaction-related anomalies (e.g., physically impossible human–object or object–object relationships) are the most frequent failure mode for current T2I models.
>  - These are followed by attribute-level anomalies (e.g., missing attachments) and then by higher-level commonsense contradictions.
>
> **[Q2] Comparison on AnomReasonor-7B and GPT-4o.**
>
> **A3:** As noted in our responses above, we believe this discrepancy is mainly due to base-model capacity rather than weaknesses in the benchmark. In the revision, we fine-tune a stronger base model (Qwen3-VL-8B with LoRA) under the same training protocol(Appendix B.5 in updated manuscript).
>
>
> **[Q3] SFT hyperparameters.**
>
> **A4:** We appreciate this question and agree that it is important to make the trade-offs transparent. The hyperparameters are shown in Appendix C. Inadditionly, we conduct a brief investigation involving three distinct training configurations for AnomReasonor-7B. Among these, Protocol B is selected in the main paper due to its optimal trade-off between classification accuracy and the quality of semantic explanations.
>
> | Protocol            | Acc | CSemAPPhe | CSemAPRea | CSemAPFull | CSemF1Phe | CSemF1Rea | CSemF1Full |
> |------------------|--------------|----------|----------|----------|----------|----------|----------|
> | Protocol A  | 77.99        | 0.2996   | 0.2833   | 0.2894   | 0.3288   | 0.3111   | 0.3178   |
> | Protocol B  | 82.61        | 0.3684   | 0.3574   | 0.3613   | 0.4048   | 0.3929   | 0.3972   |
> | Protocol C  | 70.53        | 0.4322   | 0.4243   | 0.4257   | 0.4715   | 0.4630   | 0.4645   |
>
>  Thank you once again for your valuable suggestions, which have greatly helped us enhance the clarity of our work.

---

### Official Review · Reviewer_NTPW · 2025-10-28

**Soundness:** 3
**Presentation:** 3
**Contribution:** 3
**Rating:** 6
**Confidence:** 5

**Summary:**

This paper proposes **AnomAgent**, a multi-agent system capable of automatically annotating perceptible anomalies in images, effectively replacing the need for manual labeling. Furthermore, the authors construct a benchmark containing over 20,000 samples generated through this fully automated annotation pipeline, featuring a structured output format. In the experiments, the authors validate the effectiveness of their approach from three complementary perspectives.

**Strengths:**

1. The fully automated annotation pipeline effectively addresses the inconsistency issues that commonly arise among human annotators, greatly facilitating the process of dataset construction.
2. The constructed benchmark is highly valuable for research on interpretable image forgery analysis.
3. The paper introduces the first metric designed to evaluate **semantic anomalies**, filling an important gap in semantic-level assessment for image authenticity evaluation.

**Weaknesses:**

1. It is unclear what role the **Severity Score** plays in the overall framework — please clarify its purpose and how it contributes to the evaluation or annotation process.
2. Please provide more detailed information about the **manual verification** procedure. Specifically, explain how high-quality and image-relevant annotations are selected and validated, and justify the rationality and reliability of this filtering process.

**Questions:**

Please refer to the *Weakness* section for detailed explanations.
If the authors can adequately address my concerns, I would be very willing to raise my score.

---

> ### Author Response · Authors · 2025-11-24
> **Response to Reviewer NTPW**
>
> We thank you for the clear and constructive review. We are very grateful for highlighting the value of (i) the fully automated annotation pipeline, (ii) the benchmark for interpretable image forgery analysis, and (iii) the semantic anomaly metric.
>
>
> **[W1]. Role of the Severity Score in the framework.**
>
> **A1:** Thank you for pointing out that the role of the Severity Score needed clarification.
>
> As discussed in Section 3.3, the Severity Score is used to rank anomalies, allowing practitioners to prioritize more severe anomalies when inspecting large volumes of AIGC content. When computing SemAP, we can use the predicted severity as a confidence score to rank anomalies, making the metric more sensitive to whether the model is confidently correct about important anomalies. Additionally, asking the VLM to justify a severity rating encourages deeper reasoning: the model must not only detect an anomaly but also argue why it is mild vs. severe, which we have found leads to richer descriptions(Section 2 in updated manuscript).
>
> **[W2] Details on the manual verification procedure and its reliability.**
>
> **A2:** We appreciate the opportunity to clarify the human-in-the-loop (HITL) process.
> We now provide more detail:
> - AnomAgent produces 174,872 structured anomaly candidates over 21,539 images.
> - We then outsource verification to a professional data annotation company. The approximate cost is $0.021 per anomaly, totaling around **$3,694**.
> - During the annotation process, each annotator is shown the image, and a single structured anomaly quadruple, and answers a single-choice question with three options: [ACCURATE / UNSURE / INACCURATE], based on whether the description correctly and specifically matches the image.
> - Only anomalies labeled ACCURATE are retained as GT-Known annotations (as defined in Section 3.2); UNSURE and INACCURATE entries are discarded.
>
> For quality control, we additionally randomly sample about 1% of the annotations for internal cross-checking. Use cross-validation among annotators; if an annotator shows a high disagreement rate or systematic errors, we request re-annotation of their subset. Given budget constraints, we could not further expand double-checking, but we believe this protocol provides a reasonable balance between scalability and reliability.
>
> Thank you once again for your valuable comments, which have been instrumental in enhancing the clarity and rigor of our work.

---

### Official Review · Reviewer_9Vob · 2025-10-30

**Soundness:** 3
**Presentation:** 3
**Contribution:** 2
**Rating:** 4
**Confidence:** 4

**Summary:**

This paper proposes AnomReason, a large-scale benchmark for semantic anomaly detection and reasoning in AI-generated images. It introduces AnomAgent, a modular multi-agent framework that detects and explains semantic inconsistencies through structured outputs (Name, Phenomenon, Reasoning, Severity). The dataset includes 21K photorealistic AIGC images with 170K verified anomalies. New BERTScore-based metrics (SemAP, SemF1) evaluate phenomenon- and reasoning-level accuracy. Experiments show that existing VLMs struggle with commonsense reasoning, while the fine-tuned AnomReasonor-7B achieves state-of-the-art performance.

**Strengths:**

1. The proposed AnomReason dataset and AnomAgent framework are well-designed and provide valuable resources for both the forgery detection and image generation communities.
2. The paper conducts extensive reasoning experiments across a wide range of VLMs and generative models, providing a comprehensive analysis of their capabilities in semantic-level reasoning, detection and generation.

**Weaknesses:**

1. The paper claims to employ an agent-based framework for benchmark construction; however, the system is essentially composed of prompt-driven GPT-4o modules, which cannot be considered true autonomous agents. The human-in-the-loop stage merely filters GPT-4o outputs rather than providing substantive corrections, thus retaining the inherent biases of GPT-4o. This design choice results in the strong performance of GPT-4o in Table 1—surpassing even newer models such as GPT-5 and GPT-o3—leading to a unfair comparison.
2. The evaluation metrics (SemAP and SemF1) rely on text similarity thresholds for anomaly matching, which can significantly influence performance outcomes across the Phe, Rea, and Full evaluation views. However, the paper does not provide a clear rationale or sensitivity analysis for the chosen threshold values, making it difficult to assess the stability and robustness of the reported results.
3. The paper conducts extensive experiments on various VLMs and generative models, but it fails to distill clear and generalizable conclusions from the results. Moreover, in the deepfake detection task, the evaluation lacks comparison with specialized forgery detection models.
4. In the deepfake detection task presented in Table 2, the fine-tuned AnomReasonor-7B achieves only 82% accuracy, performing even worse than GPT-4o’s zero-shot reasoning despite the absence of any cross-domain generalization challenge. This raises concerns about the validity and robustness of the proposed dataset and the effectiveness of its supervision strategy.
5. The paper defines a new task, semantic anomaly detection and reasoning, claiming that existing MLLM-based detectors operate only at the surface level while the proposed method captures content-level semantics. However, in the context of forgery detection, all MLLM-based approaches inherently perform high-level semantic analysis, as low-level forensic clues are beyond their perceptual capacity. While structuring semantic anomalies is beneficial, this additional task definition appears unnecessary.

**Questions:**

please see the weakness.

---

> ### Author Response · Authors · 2025-11-24
> **Response to Reviewer 9Vob (Part 1/2)**
>
> We thank you for the thoughtful and detailed review. We are grateful that you found the AnomReason dataset and AnomAgent framework well-designed and valuable, and that you appreciated the extensive reasoning experiments across VLMs and generative models.
>
> **[W1] Agent-based framework vs. prompt-driven GPT-4o modules; impact of GPT-4o bias.**
>
> **A1:** We thank the reviewer for this insightful comment.
>
> **(a) Agent-based framework vs. prompt-driven GPT-4o modules.**
>
> We agree that our original wording may have been unclear. In line with recent work on LLM-based agents[1], we use “agent-based” in a broad, system-level sense: role-specialized, LLM-driven modules that operate on structured state and communicate via intermediate messages, rather than a single monolithic LLM call.
>
> In this sense, AnomAgent is a modular multi-agent framework: ObjectPerceiver → AttributeAnalyzer → RelationReasoner → AnomalyIntegrator → AnomalyFormatter. Each component receives structured scene state (object lists, attributes, relations, anomaly candidates), transforms or augments it, and passes structured outputs to the next stage (Fig. 2). To avoid confusion, we will clarify in the revision that we adopt this widely used system-level notion of “LLM-based agents,” and explicitly describe each module as a role-specialized decision component within a multi-stage pipeline.
>
> REFERENCE:
> [1] Wang L, Ma C, Feng X, et al. A survey on large language model based autonomous agents.
>
> **(b) Regarding GPT-4o bias.**
>
> When we started this project, GPT-4o was the strongest widely available VLM. To select the backbone for AnomAgent empirically, we ran a pilot study on 218 Midjourney images, annotating each image with three methods: (i) human annotation, (ii) AnomAgent instantiated with GPT-4o, and (iii) AnomAgent instantiated with GPT-o3. We then compared the two AnomAgent variants against human annotations using an LLM-based similarity judge: the average similarity between AnomAgent(GPT-4o) and human annotations was 0.5936, while for AnomAgent(GPT-o3) it was 0.4559. This indicates that the GPT-4o-based pipeline aligns substantially better with human judgments. In contrast, the “thinking mode” of GPT-o3—its characteristic reasoning style and way of organizing explanations—tends to produce annotations whose structure and focus are less well aligned with our semantic anomaly detection and structuring requirements. Based on this analysis, we chose GPT-4o as the backbone for AnomAgent in this work.
>
> During the subsequent human-in-the-loop (HITL) stage, GPT-5 was released. We evaluated GPT-5 and found that, although it has stronger general reasoning ability than GPT-o3, its reasoning style still noticeably affects annotation patterns and is not fully aligned with our established protocol. For consistency of the dataset and to avoid mixing annotation regimes mid-pipeline, we therefore kept GPT-4o for annotation and used GPT-5 (and GPT-o3) only as evaluation baselines.
>
>
> **[W2] SemAP / SemF1 dependence on similarity thresholds; need for rationale and sensitivity analysis**
>
> **A2:** Thank you for your valuable feedback. We agree that the choice of thresholds is important. In the update manuscript (Appendix B.4), we provide a sensitivity analysis for SemAP and SemF1 using different threshold(Fig. 10). It can be observed that our chosen thresholds $\{0.7,0.8,0.9\}$ therefore capture three regimes, stable (0.7), stricter (0.8), and near-failure (0.9), providing a balanced measure of semantic alignment.

---

> ### Author Response · Authors · 2025-11-24
> **Response to Reviewer 9Vob (Part 2/2)**
>
> **[W3] Lack of distilled general conclusions and absence of specialized forgery detectors in comparisons.**
>
> **A3:** Thank you for your insightful feedback, which has allowed us to refine the presentation of our findings.
>
> **(a) Clearer conclusions.**
>
> Sections 4 already contain several qualitative observations, but we agree that these can be made more explicit and synthesized into clearer, high-level takeaways. In the updated manuscript, we sharpen and highlight the main conclusions as follows:
>
>  - Current VLMs exhibit limited semantic anomaly
> competence in AIGC images, particularly in reasoning. Across VLMs, detecting that “something is wrong” (Phe) is generally easier than explaining “why” it is wrong (Rea), and this gap can be substantial.
>  - Fine-tuning on AnomReason consistently reduces this gap, leading to a more balanced Phe/Rea profile and higher-quality, structured reasoning.
>  - In explainable deepfake detection, models with stronger semantic reasoning provide better-aligned anomaly explanations even when their classification accuracies are similar. Semantic reasoning provides capabilities complementary to specialized forensic detectors, rather than competing with them.
>
> We state these points in Section 4 and use them to frame the empirical results, so that readers can more easily grasp the general conclusions of our study.
>
> **(b) Comparisons to forgery detectors.**
>
> Following your suggestion, we additionally re-implement two specialized forgery detection methods (e.g., NPR and UniFD) on AnomReason-Deepfake: NPR and UniFD achieve 95.2 and 97.5 accuracy. However, those methods are unable to perform semantic anomaly reasoning effectively. In the proposed benchmark, semantic anomaly reasoning constitutes a critical evaluation aspect.
>
>
> **[W4] Performance of AnomReasonor-7B and GPT-4o.**
>
> **A4:** We agree that this difference deserves discussion. We believe it is primarily due to base-model capacity, rather than flaws in the dataset or supervision scheme. To verify this, we fine-tuned a stronger base model (Qwen3-VL-8B with LoRA adapters) using the same training protocol.
>
> | Model            | Acc | CSemAPPhe | CSemAPRea | CSemAPFull | CSemF1Phe | CSemF1Rea | CSemF1Full |
> |------------------|--------------|----------|----------|----------|----------|----------|----------|
> | GPT-4o         | 87.76        | 0.3750  | 0.3487   | 0.3612   | 0.4054   | 0.3770   | 0.3905   |
> | AnomReasonor(Qwen2.5-7B)  | 82.61        | 0.3684   | 0.3574   | 0.3613   | 0.4048   | 0.3929   | 0.3972   |
> | AnomReasonor(Qwen3-8B)  | **90.49**        | **0.3890**  | **0.3802**   | **0.3832**   | **0.4129**   | **0.4034**   | **0.4066**   |
>
> We report these results in Appendix B.5. They show that with a stronger backbone, AnomReasonor can surpass GPT-4o both in accuracy and semantic metrics, supporting the effectiveness of our supervision.
>
> **[W5]. Necessity of defining a separate “semantic anomaly detection and reasoning” task.**
>
> **A5:** Thank you for giving us the opportunity to clarify our terminology. In our work, **“semantic anomaly”** does not mean “high-level semantics,” but specifically refers to:
> > logical contradictions, physical implausibilities, and commonsense violations
> that compromise the plausibility or authenticity of a scene.
>
> Our modular multi-agent pipeline (AnomAgent) explicitly analyzes object attributes and inter-object relations to extract such semantic anomalies. Although current VLMs utilize high-level image features as visual tokens, their performance in semantic anomaly detection and reasoning tasks remains suboptimal, as demonstrated in Tables 1–2.
>
> We will rephrase the introduction to position our task as a refinement and formalization of semantic-level visual anomaly analysis: we focus on explicitly structuring the anomalies (what, why, how severe), which enables more interpretable evaluation and training.
>
> Thank you once again for your valuable comments. We hope our response can address your concerns effectively.

---

> > ### Comment · Reviewer_9Vob · 2025-11-26
> >
> > Thank you for the authors’ response. My main concerns have been addressed, and I have therefore raised my score.

---

### Official Review · Reviewer_D51R · 2025-11-02

**Soundness:** 3
**Presentation:** 3
**Contribution:** 3
**Rating:** 6
**Confidence:** 4

**Summary:**

This paper presents a novel task and dataset for semantic visual anomaly detection and reasoning, called AnomReason. The dataset was constructed using an iterative hybrid annotation framework, AnomAgent, which combines VLM generations with human verification.
The authors conduct extensive benchmarking of both open-weight and proprietary models on their dataset. Furthermore, they fine-tune a VLM on the AnomReason training set, demonstrating notable performance improvements as a result.

**Strengths:**

- The paper introduces a novel dataset that addresses an important gap in semantic visual anomaly detection and reasoning.

- The dataset’s design is rigorously supported by large-scale human annotation efforts, ensuring the validity and reliability of the resource.

- The authors provide extensive benchmarking of various vision-language models, offering valuable insights into model performance on this new task.

**Weaknesses:**

**[W1]** The paper does not provide statistics on the diversity of visual phenomena present in the benchmark. This omission makes it difficult to assess how the dataset splits are affected, particularly regarding potential entity overlap. Without this information, there is a risk of data leakage across splits in terms of topic, domain, or subject, which could impact the validity of fine-tuning and evaluation.

**[W2]** A significant limitation in the experimental results (Sections 4.1 and 4.2) is the lack of counterfactual examples, i.e. AI-generated images that do not contain semantic anomalies, or real-world images. Including such examples would allow for a more thorough assessment of the fine-tuned model’s ability to distinguish between anomalous and non-anomalous cases, and between AIGC and non-AIGC.

**[W3]** The authors rely on BERTScore to compare extracted anomalies to the ground truth. Given the known limitations of BERTScore (especially its insensitivity to subtle semantic differences) it may be more appropriate to use an LLM-based judge for evaluation, as suggested in recent literature. Also, it is important to validate the proposed metrics with human evaluation. Conducting a sampled error analysis would help identify failure modes and provide additional support for the validity of the chosen metrics.

**Questions:**

Please refer to the weaknesses.

---

> ### Author Response · Authors · 2025-11-24
> **Response to Reviewer D51R**
>
> We thank you for the thoughtful and constructive review. We appreciate your recognition that AnomReason fills an important gap in semantic visual anomaly detection and that our benchmarking is extensive.
>
> **[W1] Lack of statistics on the diversity of visual phenomena; risk of entity/topic leakage across splits.**
>
> **A1:** Thank you for highlighting this important point. In the updated manuscript, we report statistics on the diversity of semantic visual anomalies in AnomReason in Section 3.2. For dataset splitting, as described in Section 3.2, we first pool all 21,533 images and then perform a random 50%/50% split into 10,765 training and 10,774 test images. Under this standard random splitting, there is inevitably phenomenon overlap between train and test. We show that no particular category or domain is over-represented in one split.
>
> To further alleviate concerns about data leakage and overfitting, we added scaling experiments where AnomReasonor is trained on only 1/4, and 1/2 of the training set, while keeping the same test set. We observe that smooth performance scaling with more training data and stable relative rankings of models and metrics.
>
> | Training Num     | Acc   | CSemAPPhe| CSemAPRea|CSemAPFull| CSemF1Phe| CSemF1Rea| CSemF1Full|
> |------------------|-------|----------|----------|----------|----------|----------|----------|
> | Quarter          | 74.03 | 0.1775   | 0.1692   | 0.1711   | 0.2008   | 0.1912   | 0.1936   |
> | Half             | 79.63 | 0.2681   | 0.2575   | 0.2600   | 0.2920   | 0.3040   | 0.2950   |
> | Full             | 82.61 | 0.3684   | 0.3574   | 0.3613   | 0.4048   | 0.3929   | 0.3972   |
>
>
>
> **[W2] Lack of counterfactual examples.**
>
> **A2:** We appreciate this suggestion. Our original formulation already involves real images through AnomReason-Deepfake, but we agree that the role of counterfactual cases was not clearly explained.
> In the updated manuscript (Appendix B), we have incorporated a demonstration of results on real-world images, as illustrated in Figure 8-9. We adopt a two-round conversation protocol for detection and reasoning (Appendix C.2). The model first predicts whether an image is real vs. AI-generated, and conditioned on this prediction, it then produces a structured explanation. When the model predicts real, the second step defaults to a content-based description of what is observed in the image, without marking anomalies. This effectively introduces counterfactual behavior on real images and helps assess whether the model avoids hallucinating anomalies where none exist.
>
>
> **[W3] BERTScore limitations; use of LLM judges and human evaluation**
>
> **A3:** We appreciate your suggestion to involve LLM-based and human evaluation. In response, we have made the following additions:
>
> **1. LLM-based judge for semantic matching.**
>
> We replace BERTScore with a LLM-based judge in a new evaluation variant. Due to time constraints,   we employ Qwen2.5-3B as the judge. The judge compares predicted vs. ground-truth    Phenomenon/Reasoning texts and produces a normalized similarity score in [0,1]. Using thresholds τ = {0.4, 0.6, 0.8}, we compute the same CSemAP/CSemF1 metrics as before. These results are consistent with the trends seen under BERTScore, but they provide finer discrimination on subtle reasoning differences.
> Since calculating AP requires performing a greedy matching process between predictions and annotations for each sample, evaluating the test set requires invoking LLMs on a scale of millions of iterations. We therefore report a representative LLM-judge-based table:
>
>
> | Eval Model | Acc | CSemAP_Phe | CSemAP_Rea | CSemAP_Full | CSemF1_Phe | CSemF1_Rea | CSemF1_Full |
> |----------  |--------------|----------|----------|----------|----------|----------|----------|
> | BERTScore  | 82.61        | 0.3684   | 0.3574   | 0.3613   | 0.4048   | 0.3929   | 0.3972   |
> | LLM judges | 82.61    | 0.1432  | 0.2011 | 0.1257   | 0.1365  | 0.1916 | 0.1198   |
>
> **2. Human evaluation to validate metrics.**
>
>    To validate the automatic metrics, we randomly select 30 pairs of (predicted Phenomenon, ground-truth Phenomenon) and obtain three similarity scores for each pair: BERTScore, LLM-judge score, and the average of three human ratings (0–1 scale).
>  Averaging over all pairs, we obtain mean scores of 0.8537 (BERTScore), 0.6600 (LLM-judge), and 0.5556 (human). While this is a small sample, the LLM-judge’s mean is closer to the human mean than BERTScore’s, suggesting better alignment with human judgment on these pairs. We will add an additional LLM-judge-based table in the revision due to time limitation.
>
> Thank you once again for your valuable comments and suggestions, which have greatly helped us refine and improve our work.

---

### Author Response · Authors · 2025-12-03
**Review and Reviewer-Author Discussion Summary**

Dear AC, SACs, PCs, and Reviewers,

Thank you very much for the time and effort you have devoted to reviewing our submission. To assist the newly assigned AC and reduce their workload, we provide below a concise summary of the key points from the reviews and the subsequent reviewer–author discussion.

---

**Overall Assessment:** All four reviewers evaluated the paper positively in the initial round or discussion:

- **Reviewer D51R:** **6** (marginally above acceptance)
- **Reviewer 9Vob:** 4 **→ 6 after discussion**
- **Reviewer NTPW:** **6**
- **Reviewer EDEz:** **6**

After the rebuttal and discussion, Reviewer **9Vob** explicitly stated that most concerns had been addressed and raised their score from 4 to 6. The other reviewers did not express remaining blocking issues.

---

**Concerns and Our Addressing.** Below we briefly list the main concerns and where they are addressed in the  rebuttal and updated manuscript:
- **Dataset diversity & leakage (D51R)** – Added anomaly/phenomenon statistics and clarified random split in **Sec. 3.2**, plus data-scaling experiments (1/4, 1/2, full) in **App. B.3 (Table 4)**.
- **Counterfactuals images (D51R)** – Clarified the role of real images and the two-round protocol in **App. C.2**, and added real-image examples in **App. B (Figs. 8–9)**.
- **Metrics: BERTScore, thresholds, human validation (D51R, 9Vob)** – Introduced an **LLM-based judge**, added threshold sensitivity for SemAP/SemF1 in **App. B.4 (Fig. 10)**, and reported a small human study.
- **Agent-based terminology & GPT-4o bias (9Vob)** – Clarified the modular agent design in **Sec. 3.1**, reported a 218-image pilot comparing GPT-4o vs. GPT-o3, and added stronger-backbone results **AnomReasonor(Qwen3-8B)** in **App. B.5**.
- **Scope, forgery detectors, and model novelty (9Vob, EDEz)** – Emphasized benchmark/task/pipeline/metrics as main contributions, positioned AnomReasonor as a simple baseline, added NPR/UniFD results on AnomReason-Deepfake and an SFT ablation (Protocols A/B/C).
- **Severity Score & HITL reliability (NTPW)** – Clarified the role of Severity in **Sec. 2/3.3** and expanded the HITL verification protocol (labels, cost, 1% spot-check, re-annotation).
---
**Main Strengths.** Across the four reviews, several core strengths consistently emerged:

1. **Novel benchmark and task for semantic visual anomaly detection and reasoning in AIGC images with structured quadruple annotations.**
   → Acknowledged by D51R:Strengths 1, 9Vob:Strengths 1, NTPW:Strengths 2, EDEz:Strengths 2.
2. **Scalable annotation pipeline and semantic metrics, via the AnomAgent multi-agent framework, HITL verification, and SemAP/SemF1 for anomaly-level evaluation.**
   → Acknowledged by D51R:Strengths 2, 9Vob:Strengths 1, NTPW:Strengths 1&3, EDEz:Strengths 2.
3. **Extensive experiments across diverse VLMs and generative models**.
   → Acknowledged by D51R:Strengths 3, 9Vob:Strengths 2.

---

**Reviewer Recognition After Discussion:**
- Reviewer **9Vob** explicitly acknowledged that most concerns had been addressed and increased their score from **4 to 6**.
- The other three reviewers (**D51R, NTPW, EDEz**) already had scores of 6 and did not raise additional blocking issues after the rebuttal.

---

We hope this summary accurately reflects the reviewers’ comments and our responses, and that it assists the AC in forming a final judgment. We are deeply grateful to the reviewers, AC, SAC, and PC for their time and insightful feedback, which have substantially strengthened the paper.

*Sincerely,*
*The Authors*

---

### Meta-Review · Area_Chair_3AVk · 2026-01-05

**Summary:**

This paper proposes a new dataset for semantic visual anomaly detection and reasoning. All reviewers find the studied setting novel and the results provide new insights. The authors’ rebuttal has successfully addressed the major concerns of reviewers. Overall, I recommend acceptance of this submission.

In particular, based on my understanding, the following primary concerns from the initial version have been properly addressed:

- Metrics abd Valudation - more explanations and additional results
- Presentation and clarity of technical novelty
- New analysis based on reviewers' requests

**Reviewer Concerns:**

All major concerns are addresed

**Reviewer Scores:**

The reviewer who gave 4 already agreed to increase to 6

---

### Decision · Program_Chairs · 2026-01-26

Accept (Poster)